# Hierarchical Open-vocabulary Universal Image Segmentation

Xudong Wang[1*]    Shufan Li[1*]    Konstantinos Kallidromitis[2*]
Yusuke Kato[2]    Kazuki Kozuka[2]    Trevor Darrell[1]
[1]Berkeley AI Research, UC Berkeley      [2]Panasonic AI Research
project page: http://people.eecs.berkeley.edu/~xdwang/projects/HIPIE

## Abstract

Open-vocabulary image segmentation aims to partition an image into semantic regions according to arbitrary text descriptions. However, complex visual scenes can be naturally decomposed into simpler parts and abstracted at multiple levels of granularity, introducing inherent segmentation ambiguity. Unlike existing methods that typically sidestep this ambiguity and treat it as an external factor, our approach actively incorporates a hierarchical representation encompassing different semantic-levels into the learning process. We also propose a decoupled text-image fusion mechanism and representation learning modules for both "things" and "stuff".[1] Additionally, we systematically examine the differences that exist in the textual and visual features between these types of categories. Our resulting model, named **HIPIE**, tackles **HI**erarchical, o**P**en-vocabulary, and un**I**v**E**rsal segmentation tasks within a unified framework. Benchmarked on over 40 datasets, *e.g.*, ADE20K, COCO, Pascal-VOC Part, RefCOCO/RefCOCOg, ODinW and SeginW, HIPIE achieves the state-of-the-art results at various levels of image comprehension, including semantic-level (*e.g.*, semantic segmentation), instance-level (*e.g.*, panoptic/referring segmentation and object detection), as well as part-level (*e.g.*, part/subpart segmentation) tasks.

## 1   Introduction

Image segmentation is a fundamental task in computer vision, enabling a wide range of applications such as object recognition, scene understanding, and image manipulation [50, 14, 42, 7, 37]. Recent advancements in large language models pave the way for open-vocabulary image segmentation, where models can handle a wide variety of object classes using text prompts. However, there is no single "correct" way to segment an image. The inherent ambiguity in segmentation stems from the fact that the interpretations of boundaries and regions within an image depend on the specific tasks.

Existing methods for open-vocabulary image segmentation typically address the ambiguity in image segmentation by considering it as an external factor beyond the modeling process. In contrast, we adopt a different approach by embracing this ambiguity and present **HIPIE**, as illustrated in Fig. 1, a novel **HI**erarchical, o**P**en-vocabulary and un**I**v**E**rsal image segmentation and detection model. This includes semantic-level segmentation, which focuses on segmenting objects based on their semantic meaning, as well as instance-level segmentation, which involves segmenting individual instances of objects or groups of objects (*e.g.*, instance and referring segmentation).

---

[1]The terms *things* (countable objects, typically foreground) and *stuff* (non-object, non-countable, typically background) [1] are commonly used to distinguish between objects that have a well-defined geometry and are countable, *e.g.* people, cars, and animals, and surfaces or regions that lack a fixed geometry and are primarily identified by their texture and/or material, *e.g.* the sky, road, and water body.

*: equal contribution

37th Conference on Neural Information Processing Systems (NeurIPS 2023).

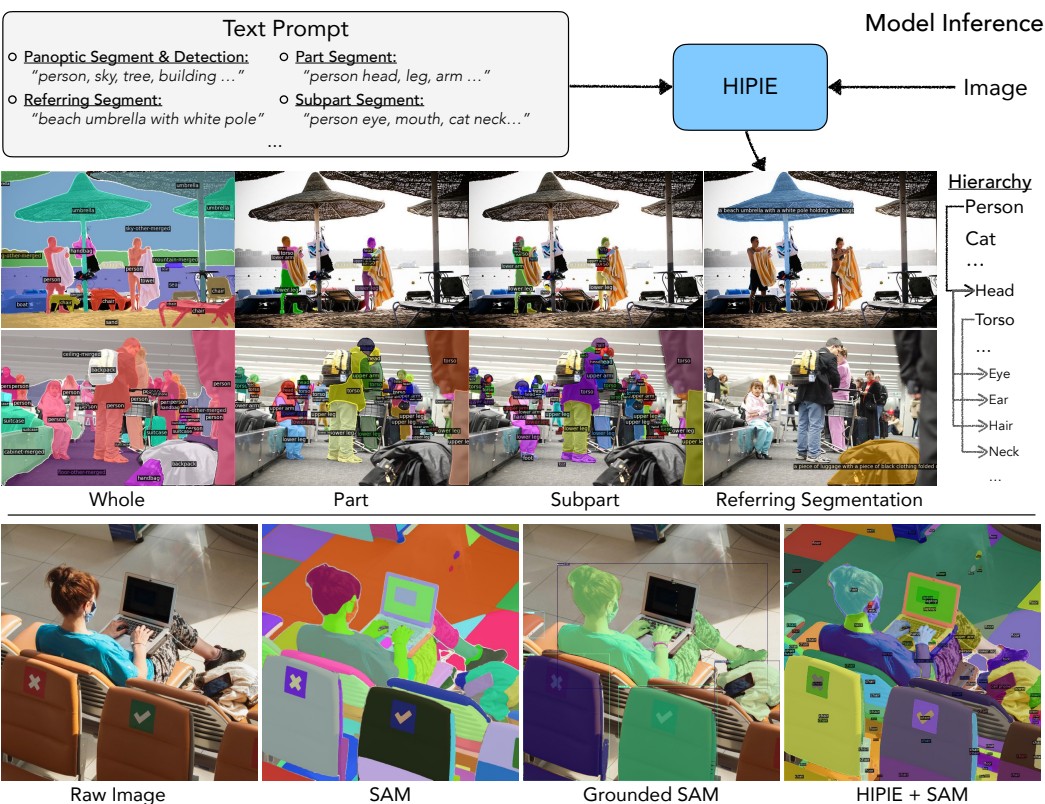

**Figure 1:** HIPIE is a unified framework which, given an image and a set of arbitrary text descriptions, provides hierarchical semantic, instance, part, and subpart-level image segmentations. This includes open-vocabulary semantic (*e.g.*, crowds and sky), instance/panoptic (*e.g.*, person and cat), part (*e.g.*, head and torso), subpart (*e.g.*, ear and nose) and referring expression (*e.g.*, umbrella with a white pole) masks. HIPIE outperforms previous methods and established new SOTAs on these tasks regardless of their granularity or task specificity. Bottom images: our method can seamlessly integrate with SAM to enable class-aware image segmentation on SA-1B.

Additionally, our model captures finer details by incorporating part-level segmentation, which involves segmenting object parts/subparts. By encompassing different granularity, HIPIE allows for a more comprehensive and nuanced analysis of images, enabling a richer understanding of their contents.

To design HIPIE, we begin by investigating the design choices for open-vocabulary image segmentation (OIS). Existing methods on OIS typically adopt a text-image fusion mechanism, and employ a shared representation learning module for both stuff and thing classes [4, 62, 58, 10, 56]. Fig. 2 shows the similarity matrics of visual and textual features between stuff and thing classes. On this basis, we can derive several conclusions:

- Noticeable discrepancies exist in the between-class similarities of textual and visual features between stuff and thing classes.
- Stuff classes exhibit significantly higher levels of similarity in text features than things.

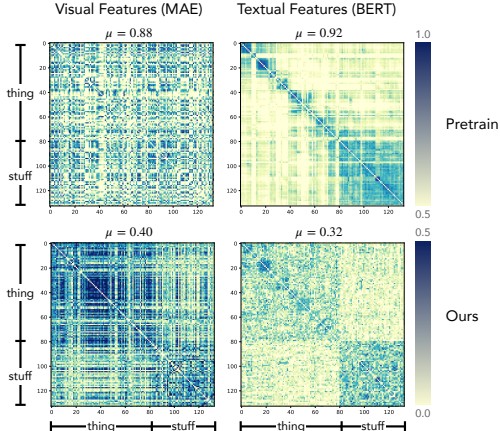

**Figure 2:** Noticeable discrepancies exist in the between-class similarities of visual and textual features between stuff and thing classes. We propose a decoupled representation learning approach that effectively generates more discriminative visual and textual features. We extract similarity matrices for the visual features, obtained through a pretrained MAE [17] or our fine-tuned one, and for the text features, produced using a pretrained BERT [6] or fine-tuned one. We report results on COCO-Panoptic [23] and measure the mean similarity ($\mu$).

|  | Open Vocab. | Instance Segment. | Semantic Segment. | Panoptic Segment. | Referring Segment. | Cls-agnostic Part Seg. | Cls-aware Part Seg. | Object Detection |
|---|---|---|---|---|---|---|---|---|
| SAM [24] | ✗ | ✓ | ✗ | ✗ | ✗ | ✓ | ✗ | * |
| SEEM [67] | ✓ | ✓ | ✓ | ✓ | ✓ | ✗ | ✗ | * |
| ODISE [56] | ✓ | ✓ | ✓ | ✓ | ✗ | ✗ | ✗ | * |
| UNINEXT [58] | † | ✓ | ✗ | ✗ | ✓ | ✗ | ✗ | ✓ |
| X-Decoder [66] | ✓ | ✓ | ✓ | ✓ | ✓ | ✗ | ✗ | * |
| G-DINO [36] | ✓ | ✓ | ✗ | ✗ | ✓ | ✗ | ✗ | ✓ |
| PPS [5] | ✗ | ✗ | ✗ | ✗ | ✗ | ✓ | ✓ | ✗ |
| HIPIE | ✓ | ✓ | ✓ | ✓ | ✓ | ✓ | ✓ | ✓ |
| *vs. prev. SOTA* | - | **+5.1** | **+2.0** | **+1.3** | **+0.5** | - | **+5.2** | **+3.2** |

**Table 1:** Our HIPIE is capable of performing all the listed segmentation and detection tasks and achieves the state-of-the-art performance using a unified framework. We present performance comparisons with SOTA methods on a range of benchmark datasets: $AP^{mask}$ for instance segmentation on MSCOCO [34], $AP^{box}$ for object detection on MSCOCO, oIoU for referring segmentation on RefCOCO+ [61], mIoU for semantic segmentation on Pascal Context[64], and $mIoU_{PartS}$ for part segmentation on Pascal-Panoptic-Parts [5]. The second best performing method for each task is underlined. *: object detection can be conducted via generating bounding boxes using instance segmentation masks. 'Seg.' denotes segmentation. †: In principle, UNINEXT can take arbitrary texts as labels, however, the original work focused on close-set performance and did not explore open-vocabulary inference.

This observation suggests that integrating textual features may yield more significant benefits in generating discriminative features for thing classes compared to stuff classes. Consequently, for thing classes, we adopt an early image-text fusion approach to fully leverage the benefits of discriminative textual features. Conversely, for stuff classes, we utilize a late image-text fusion strategy to mitigate the potential negative effects introduced by non-discriminative textual features. Furthermore, the presence of discrepancies in the visual and textual features between stuff and thing classes, along with the inherent differences in their characteristics (stuff classes requiring better capture of texture and materials, while thing classes often having well-defined geometry and requiring better capture of shape information), indicates the need for decoupling the representation learning modules for producing masks for stuffs and things.

In addition to instance/semantic-level segmentation, our model is capable of open-vocabulary hierarchical segmentation. Instead of treating part classes, like 'dog leg', as standard multi-word labels, we concatenate class names from different granularity as prompts. During training, we supervise the classification head using both part labels, such as 'tail', and instance labels, such as 'dog', and we explicitly contrast a mask embedding with both instance-level and part-level labels. In the inference stage, we perform two separate forward passes using the same image but different prompts to generate instance and part segmentation. This design choice empowers *open-vocabulary* hierarchical segmentation, allowing us to perform part segmentation on novel part classes by randomly combining classes from various granularity, such as 'giraffe' and 'leg', even if they have never been seen during training. By eliminating the constraints of predefined object classes and granularity, HIPIE offers a more flexible and adaptable solution for image segmentation.

We extensively benchmark HIPIE on various popular datasets to validate its effectiveness, including MSCOCO, ADE20K, Pascal Panoptic Part, and RefCOCO/RefCOCOg. HIPIE achieves state-of-the-art performance across all these datasets that cover a variety of tasks and granularity.

To the best of our knowledge, HIPIE is the first hierarchical, open-vocabulary and universal image segmentation and detection model (see Table 1). By decoupling representation learning and text-image fusion mechanisms for things vs. stuff classes, HIPIE overcomes the limitations of existing approaches and achieves state-of-the-art performance on various benchmarks.

## 2   Related Works

**Open-Vocabulary Semantic Segmentation** [2, 53, 26, 16, 44, 32, 54, 55] aims to segment an image into semantic regions indicated by text descriptions that may not have been seen during training. ZS3Net [2] combines a deep visual segmentation model with an approach to generate

visual representations from semantic word embeddings to learn pixel-wise classifiers for novel categories. LSeg [26] uses CLIP's text encoder [43] to generate the corresponding semantic class's text embedding, which it then aligns with the pixel embeddings. OpenSeg [16] adopts a grouping strategy for pixels prior to learning visual-semantic alignments. By aligning each word in a caption to one or a few predicted masks, it can scale-up the dataset and vocabulary sizes. GroupViT [54] is trained on a large-scale image-text dataset using contrastive losses. With text supervision alone, the model learns to group semantic regions together. OVSegmentor [55] uses learnable group tokens to cluster image pixels, aligning them with the corresponding caption embeddings.

**Open-Vocabulary Panoptic Segmentation** (OPS) unifies semantic and instance segmentation, and aims to perform these two tasks for arbitrary categories of text-based descriptions during inference time [10, 56, 66, 67, 58]. MaskCLIP [10] first predicts class-agnostic masks using a mask proposal network. Then, it refines the mask features through Relative Mask Attention interactions with the CLIP visual model and integrates the CLIP text embeddings for open-vocabulary classification. ODISE [56] unifies Stable Diffusion [47], a pre-trained text-image diffusion model, with text-image discriminative models, *e.g.* CLIP [43], to perform open-vocabulary panoptic segmentation. X-Decoder [66] takes two types of queries as input: generic non-semantic queries that aim to decode segmentation masks for universal segmentation, and textual queries to make the decoder language-aware for various open-vocabulary vision tasks. UNINEXT [58] unifies diverse instance perception tasks into an object discovery and retrieval paradigm, enabling flexible perception of open-vocabulary objects by changing the input prompts.

**Referring Segmentation** learns valid multimodal features between visual and linguistic modalities to segment the target object described by a given natural language expression [19, 60, 20, 22, 13, 59, 52, 35, 63]. It can be divided into two main categories: 1) *Decoder-fusion* based method [8, 51, 63, 35] first extracts vision features and language features, respectively, and then fuses them with a multi-modal design. 2) *Encoder-fusion* based method [13, 59, 30] fuses the language features into the vision features early in the vision encoder.

**Parts Segmentation** learns to segment instances into more fine-grained masks. PPP [5] established a baseline of hierarchical understanding of images by combining a scene-level panoptic segmentation model and part-level segmentation model. JPPF [21] improved this baseline by introducing joint Panoptic-Part Fusion module that achieves comparable performance with significantly smaller models.

**Promptable Segmentation.** The Segment Anything Model (SAM) [24] is an approach for building a fully automatic promptable image segmentation model that can incorporate various types of human interventions, such as texts, masks, and points. SEEM [67] proposes a unified prompting scheme that encodes user intents into prompts in a joint visual-semantic space. This approach enables SEEM to generalize to unseen prompts for segmentation, achieving open-vocabulary and zero-shot capabilities. Referring segmentation can also be considered as promptable segmentation with text prompts.

**Comparison with Previous Work.** Table 1 compares our HIPIE method with previous work in terms of key properties. Notably, HIPIE is the only method that supports open-vocabulary universal image segmentation and detection, enabling the object detection, instance-, semantic-, panoptic-, hierarchical-(whole instance, part, subpart), and referring-segmentation tasks, all within a single unified framework.

# 3 Method

We consider all relevant tasks under the unified framework of language-guided segmentation, which performs open-vocabulary segmentation and detection tasks for arbitrary text-based descriptions.

## 3.1 Overall Framework

The proposed HIPIE model comprises three main components, as illustrated in Fig. 3:

*1) Text-image feature extraction and information fusion (detailed in Secs. 3.2 to 3.4):* We first generate a text prompt $T$ from labels or referring expressions. Then, we extract image ($I$) and text ($T$) features $F_v = \text{Enc}_v(I), F_t = \text{Enc}_t(T)$ using image encoder $\text{Enc}_v$ and text encoder $\text{Enc}_t$, respectively. We then perform feature fusion and obtain fused features $F'_v, F'_t = \text{FeatureFusion}(F_v, F_t)$.

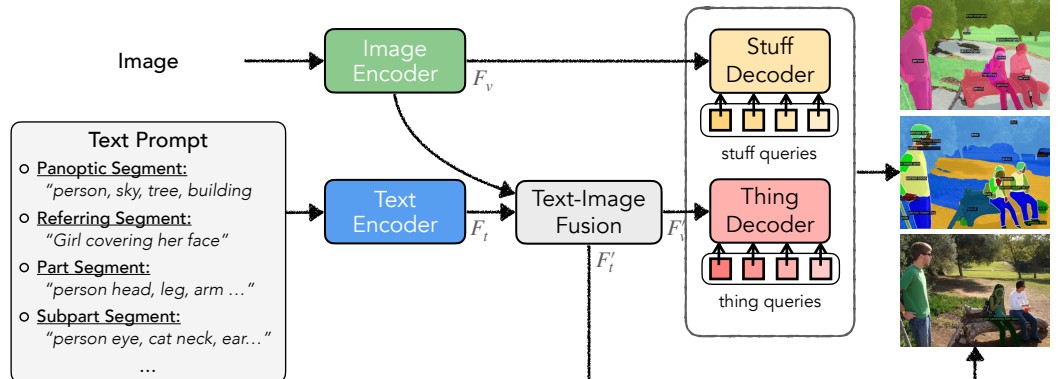

**Figure 3:** Diagram of HIPIE for hierarchical, universal and open-vocabulary image segmentation and detection. The image and text prompts are first passed to the image and text decoder to obtain visual features $F_v$ and text features $F_t$. Early fusion is then applied to merge image and text features to get $F_v'$, $F_t'$. Two independent decoders are used for things (foreground) classes and stuff (background) classes.

*2) Foreground (referred to as things) and background (referred to as stuffs) mask generation (detailed in Sec. 3.5):* Each of the decoders takes in a set of image features and text features and returns sets of masks, bounding boxes, and object embeddings $(M, B, E)$. We compute the foreground and background proposals and concatenate them to obtain the final proposals and masks as follows:

$$
\begin{aligned}
\text{Stuff}: \quad (M_2, B_2, E_2) &= \text{StuffDecoder}(F_v, F_t) \\
\text{Thing}: \quad (M_1, B_1, E_1) &= \text{ThingDecoder}(\text{FeatureFusion}(F_v, F_t)) \\
\text{Overall}: \quad (M, B, E) &= (M_1 \oplus M_2, B_1 \oplus B_2, E_1 \oplus E_2)
\end{aligned}
\tag{1}
$$

where $\oplus$ denotes the concatenation operation.

*3) Proposal and mask retrieval using text prompts (detailed in Sec. 3.6):* To assign class labels to these proposals, we compute the cosine similarity between object embedding $E$ and the corresponding embedding $E_i'$ of class $i \in \{1, 2..., c\}$. For a set of category names, the expression is a concatenated string containing all categories. We obtain $E_i'$ by pooling tokens corresponding to each label from the encoded sequence $F_t$. For referring expressions, we taken the [CLS] token from BERT output as $E_i'$.

## 3.2 Text Prompts

Text prompting is a common approach used in open-vocabulary segmentation models [19, 60, 57, 58].

For open-vocabualry instance segmentation, panoptic segmentation, and semantic segmentation, the set of all labels $C$ is concatenated into a single text prompt $T_i$ using a "." delimiter. Given an image $I$ and a set of text prompts $T$, the model aims to classify $N$ masks in the label space $C \cup \{\text{"}other\text{"}\}$, where $N$ is the maximum number of mask proposals generated by the model.

For referring expressions, the text prompt is simply the sentence itself. The goal is to locate one mask in the image corresponding to the language expression.

## 3.3 Image and Text Feature Extraction

We employ a pretrained BERT model [6] to extract features for text prompts. Because the BERT-base model can only process input sequences up to 512 tokens, we divide longer sequences into segments of 512 tokens and encode each segment individually. The resulting features are then concatenated to obtain features of the original sequence length.

We utilize ResNet-50 [18] and Vision Transformer (ViT) [11] as base architectures for image encoding. In the case of ResNet-50, we extract multiscale features from the last three blocks and denote them as $F_v$. For ViT, we use the output features from blocks 8, 16, and 32 as the multiscale features $F_v$.

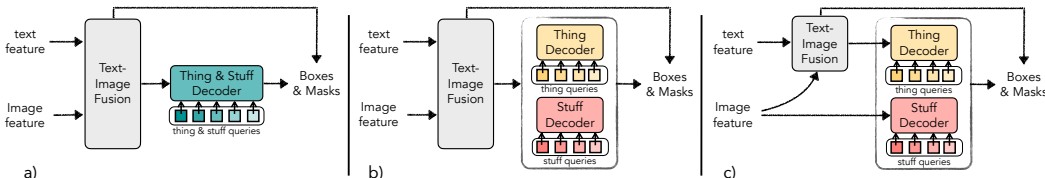

**Figure 4:** Various design choices for generating thing and stuff masks with arbitrary text descriptions. In version a), We use a single decoder for all masks. Early fusion is applied. In version b), two independent decoders are used for things and stuff classes. Early fusion is adopted for both decoders. Version c) is identical to version b) with the only difference being that the stuff decoder do not make use of early fusion.

## 3.4 Text-Image Feature Fusion

We explored several design choices for text-image feature fusion and mask generation modules as shown in Fig. 4 and Table 5, and discovered that Fig. 4c) can give us the optimal performance. We adopt bi-directional cross-attention (Bi-Xattn) to extract text-guided visual features $F_{t2v}$ and image-guided text features $F_{v2t}$. These attentive features are then integrated with the vanilla text features $F_t$ and image features $F_v$ through residual connections, as shown below:

$$
\begin{aligned}
F_{t2v},\ F_{v2t} &= \text{Bi-Xattn}(F_v, F_t) \\
(F'_v,\ F'_t) &= (F_v + F_{t2v},\ F_t + F_{v2t})
\end{aligned}
\tag{2}
$$

where $F_v$ and $F_t$ represent the visual and text-prompt features, respectively.

## 3.5 Thing and Stuff Mask Generation

We then generate masks and proposals for the thing and stuff classes by utilizing $F'_v$ and $F'_t$ that we obtained in Sec. 3.4.

**Model Architecture.** While architectures such as Mask2Former and MaskDINO [4, 28] can perform instance, semantic and panoptic segmentation simultaneously, models trained jointly show inferior performance compared with the same model trained for a specific task (*e.g.* instance segmentation only). We hypothesize that this may result from the different distribution of spatial location and geometry of foreground instance masks and background semantic masks. For example, instance masks are more likely to be connected, convex shapes constrained by a bounding box, whereas semantic masks may be disjoint, irregular shapes spanning across the whole image.

To address this issue, in a stark contrast to previous approaches [58, 36, 57] that use a unified decoder all both stuffs and things, we decouple the stuff and thing mask prediction using two separate decoders. For the thing decoder, we adopt Deformable DETR [65] with a mask head following the UNINEXT [58] architecture and incorporate denoising procedures proposed by DINO [62]. For the stuff decoder, we use the architecture of MaskDINO [28].

**Proposal and Ground-Truth Matching Mechanisms.** We make the following distinctions between the two heads. For thing decoder, we adopt simOTA [15] to perform many-to-one matching between box proposals and ground truth when calculating the loss. We also use box-iou-based NMS to remove duplicate predictions. For the stuff decoder, we adopt one-to-one Hungarian matching [25]. Additionally, we disable the box loss for stuff masks. We set the number of queries to 900 for the things and 300 for the stuffs.

**Loss Functions.** For both decoders, we calculate the class logits as the normalized dot product between mask embeddings ($M$) and text embeddings ($F'_t$). We adopt Focal Loss [33] for classification outputs, L1 loss, and GIoU loss [45] for box predictions, pixel-wise binary classification loss and DICE loss [49] for mask predictions. Given predictions $(M_1, B_1, E_1), (M_2, B_2, E_2)$, groundtruth labels $(M', B', C)$ and its foreground and background subset $(M'_f, B'_f, C_f)$ and $(M'_b, B'_b, C_b)$, The final Loss is computed as

$$
\begin{aligned}
\mathcal{L}_{\text{thing}} &= \lambda_{\text{cls}}\mathcal{L}_{\text{cls}}(E_1, C'_f) + \lambda_{\text{mask}}\mathcal{L}_{\text{mask}}(M_1, M'_f) + \lambda_{\text{box}}\mathcal{L}_{\text{box}}(B_1, B'_f) \\
\mathcal{L}_{\text{stuff}} &= \lambda_{\text{cls}}\mathcal{L}_{\text{cls}}(E_2, C') + \lambda_{\text{mask}}\mathcal{L}_{\text{mask}}(M_2, M') + \lambda_{\text{box}}\mathcal{L}_{\text{box}}(B_2, B'_b) \\
\mathcal{L} &= \mathcal{L}_{\text{thing}} + \mathcal{L}_{\text{stuff}}
\end{aligned}
\tag{3}
$$

where $\mathcal{L}_{\text{box}} = \lambda_{L1}\mathcal{L}_{L1} + \lambda_{\text{giou}}\mathcal{L}_{\text{giou}}$, $\mathcal{L}_{\text{mask}} = \lambda_{\text{ce}}\mathcal{L}_{\text{ce}} + \lambda_{\text{dice}}\mathcal{L}_{\text{dice}}$, and $\mathcal{L}_{\text{cls}} = \mathcal{L}_{\text{focal}}$. Note that while we do not use the stuff decoder for thing prediction, we still match its predictions with things and

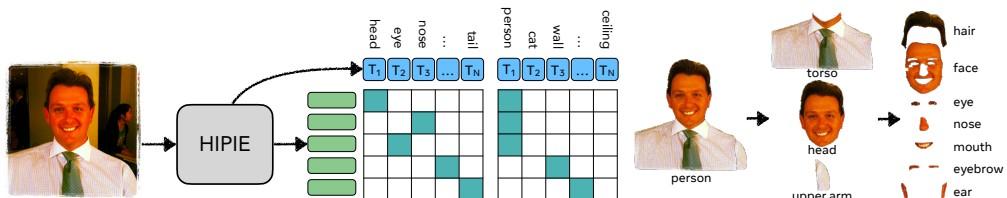

**Figure 5:** Hierarchical segmentation pipeline. We concatenate the instance class names and part class names as labels. During the training process, we supervise the classification head using both part labels and instance labels. During inference, we perform two separate forward passes using the same image but different prompts to generate instance and part segmentations. By combining the part segmentation and instance segmentation of the same image, we obtain hierarchical segmentation results on the right side.

compute the class and box losses in the training. We find such auxiliary loss setup make the stuff decoder aware of the thing distribution and imporves the final performance.

### 3.6 Open-Vocabulary Universal Segmentation

In closed set setting, we simply merge the output of two decoders and perform the standard postprocessing of UNINEXT [58] and MaskDINO [28] to obtain the final output.

In zero-shot open vocabulary setting, we follow ODISE [56] and combining our classification logits with a text-image discriminative model, *e.g.*, CLIP [43]. Specially, given the a mask $M$ on image $I$, its features $E$ and test classes $C_{\text{test}}$, we first compute the probability $p_1(E, C_{\text{test}}) = \mathbb{P}(C_{\text{test}}|E)$ in the standard way as mentioned before. We additionally compute mask-pooled features of $M$ from the vision encoder $\mathcal{V}$ of CLIP as $E_{\text{CLIP}} = \text{MaskPooling}(M, \mathcal{V}(I))$. Then we compute the CLIP logits $p_2(E, C_{\text{test}}) = \mathbb{P}(C_{\text{test}}|E_{\text{CLIP}})$ as the similarity between the CLIP text features and the $E_{\text{CLIP}}$. Finally we combine the final prediction as

$$p_{\text{final}}(E, C_{\text{test}}) \propto p_1(E, C_{\text{test}})^{\lambda} p_2(E, C_{\text{test}})^{1-\lambda} \tag{4}$$

Where $\lambda$ is a balancing factor. Emprically, we found such setting leads to better performance than naively relying completely on CLIP features only or close-set logits.

### 3.7 Hierarchical segmentation

In addition to the instance-level segmentation, we can also perform part-aware hierarchical segmentation. We concatenate the instance class names and part class names as labels. Some examples are "human ear", and "cat head". In the training process, we supervise the classification head with both part labels and instance labels. Specifically, we replace $L_{cls}$ with $L_{clsPart} + L_{clsThing}$ in Eq. (3). We combine parts segmentation and instance segmentation of the same image to get part-aware instance segmentation. Additionally, layers of hierarchy is obtained by grouping the parts. For example, the "head" consists of ears, hair, eyes, nose, etc. Fig. 5 illustrates this process. Fig. A1 highlights the difference of our approach with other methods.

### 3.8 Class-aware part segmentation with SAM

We can also perform the class-aware hierarchical segmentation by combining our semantic output with class-agnostic masks produced by SAM [24]. Specifically, given semantic masks $M$, their class probability $P_M$, and SAM-generated part masks $S$, we compute the class probability of mask $S_i \in S$ with respect to class $j$ as

$$P_S(S_i, j) \propto \sum_{M_k \in M} P_M(M_k, j)|M_k \cap S_i| \tag{5}$$

Where $|M_k \cap S_i|$ is the area of intersection between mask $M_k$ and $S_i$. We combine our semantic output with SAM because our pretraining datasets only contains object-centric masks, whereas the SA-1B dataset used by SAM contains many local segments and object parts.

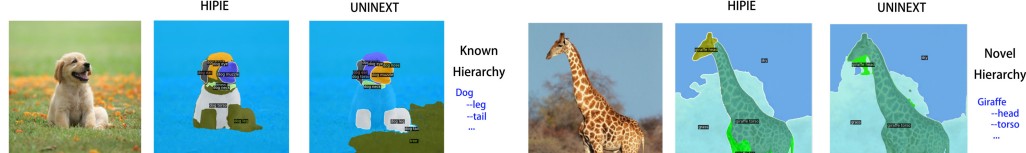

**Figure 6: Qualitative Analysis of Open Vocabulary Hierarchal Segmentation**. Because of our hierarchal design, our model produces better-quality masks. In particular, our model can generalize to novel hierarchies that do not exist in part segmentation datasets.

| Method | Backbone | COCO | | | | ADE20K | | | | PAS-P |
|--------|----------|------|------|------|------|--------|------|------|------|-------|
| | | PQ | $AP^{mask}$ | $AP^{box}$ | mIoU | PQ | $AP^{mask}$ | $AP^{box}$ | mIoU | $mIoU_{PartS}$ |
| MaskCLIP [10] | ViT16 | - | - | - | - | 15.1 | 6.0 | - | 23.7 | - |
| X-Decoder [66] | FocalT | 52.6 | 41.3 | - | 62.4 | 18.8 | 9.8 | - | 25.0 | - |
| X-Decoder | DaViT-B | 56.2 | 45.8 | - | 66.0 | 21.1 | 11.7 | - | 27.2 | - |
| SEEM [67] | FocalT | 50.6 | 39.5 | - | 61.2 | - | - | - | - | - |
| SEEM | DaViT-B | 56.2 | 46.8 | - | 65.3 | - | - | - | - | - |
| ODISE [56] | ViT-H+SD | 55.4 | 46.0 | 46.1 | 65.2 | **22.6** | 14.4 | 15.8 | **29.9** | - |
| JPPF [21] | EffNet-b5 | - | - | - | - | - | - | - | - | 54.4 |
| PPS [5] | RNST269 | - | - | - | - | - | - | - | - | 58.6 |
| HIPIE | RN50 | 52.7 | 45.9 | 53.9 | 59.5 | 18.4 | 13.0 | 16.2 | 26.8 | 57.2 |
| HIPIE | ViT-H | **58.0** | **51.9** | **61.3** | **66.8** | 20.6 | **15.0** | **18.7** | 29.0 | **63.8** |

**Table 2:** Open-vocabulary panoptic segmentation (PQ), instance segmentation ($AP^{mask}$), semantic segmentation (mIoU), part segmentation ($mIoU_{PartS}$), and object detection ($AP^{box}$). N/A: not applicable. -: not reported.

## 4  Experiments

We comprehensively evaluate HIPIE through quantitative and qualitative analyses to demonstrate its effectiveness in performing various types of open-vocabulary segmentation and detection tasks. The implementation details of HIPIE are explained in Sec. 4.1. Sec. 4.2 presents the evaluation results of HIPIE. Additionally, we conduct an ablation study of various design choices in Sec. 4.3.

### 4.1  Implementation Details

**Model Learning Settings** can be found in our appendix materials.

**Evaluation Metrics.** *Semantic Segmentation* performance is evaluated using the mean Intersection-Over-Union (mIoU) metric. For ***Part segmentation,*** we report $mIoU_{PartS}$, which is the mean IoU for part segmentation on grouped part classes [5]. ***Object Detection and Instance Segmentation*** results are measured using the COCO-style evaluation metric - mean average precision (AP) [34]. ***Panoptic Segmentation*** is evaluated using the Panoptic Quality (PQ) metric [23]. ***Referring Image Segmentation*** (RIS) [19, 60] is evaluated with overall IoU (oIoU).

### 4.2  Results

**Panoptic Segmentation.** We examine Panoptic Quality (PQ) performance across MSCOCO [34] for closed-set and ADE20K [64] for open-set zero shot transfer learning. Based on Table 3 our model is able to outperform the previous close-set state-of-the-art using a ViT-H backbone by +1.8. In addition, we match the best open-set PQ results, while being able to run on more tasks and having a simpler backbone than ODISE [56]. **Semantic Segmentation.** The evaluation of our model's performance on various open-vocabulary semantic segmentation datasets is presented in Table 4. These datasets include: 1) A-150: This dataset comprises 150 common classes from ADE20K [64]. 2) A-847: This dataset includes all 847 classes from ADE20K [64]. 3) PC-59: It consists of 59 common classes from Pascal Context [39]. 4) PC-459: This dataset encompasses the full 459 classes of Pascal Context [39]. 5) PAS-21: The vanilla Pascal VOC dataset [12], containing 20 foreground classes and 1 background class. These diverse datasets enable a comprehensive evaluation of our model's performance across different settings, such as varying class sizes and dataset complexities. Table 4 provides insights into how our model performs in handling open-vocabulary semantic segmentation tasks, demonstrating

| Method | Data | A-150 | | | | A-847 | CTX459 | SeginW |
|---|---|---|---|---|---|---|---|---|
| | | PQ | AP$^{mask}$ | AP$^{box}$ | mIoU | mIoU | mIoU | AP$^{mask}$ |
| OpenSeed | O365,COCO | 19.7 | 15.0 | 17.7 | 23.4 | - | - | 36.1 |
| X-Decoder | COCO,CC3M,SBU-C,VG,COCO-Caption,(Florence) | 21.8 | 13.1 | - | 29.6 | 9.2 | 16.1 | 32.2 |
| UNINEXT | O365,COCO,RefCOCO | 8.9 | 14.9 | 11.9 | 6.4 | 1.8 | 5.8 | 42.1 |
| HIPIE w/o CLIP | O365,COCO,RefCOCO,PACO | 18.1 | 16.7 | 20.2 | 19.8 | 4.8 | 12.2 | 41.0 |
| HIPIE w/ CLIP | + (CLIP) | 22.9 | 19.0 | 22.9 | 29.0 | 9.7 | 14.4 | 41.6 |

**Table 3:** Open-Vocabulary Universal Segmentation. We compare against other universal multi-task segmentation models. (*) denotes pretraining dataset of representations.

| Method | A-150 | PC-59 | PAS-21 | COCO |
|---|---|---|---|---|
| ZS3Net [2] | - | 19.4 | 38.3 | - |
| LSeg+ [26, 16] | 18.0 | 46.5 | - | 55.1 |
| HIPIE | 26.8 | 53.6 | 75.7 | 59.5 |
| *vs. prev. SOTA* | **+7.1** | **+10.7** | **+28.3** | **+4.4** |
| GroupViT [54] | 10.6 | 25.9 | 50.7 | 21.1 |
| OpenSeg [16] | 21.1 | 42.1 | - | 36.1 |
| MaskCLIP [10] | 23.7 | 45.9 | - | - |
| ODISE [56] | 29.9 | 57.3 | 84.6 | 65.2 |
| HIPIE | 29.0 | 59.3 | 83.3 | 66.8 |
| *vs. prev. SOTA* | -0.9 | **+2.0** | -1.3 | **+1.6** |

**Table 4:** Comparison on open-vocabulary semantic segmentation. Baseline results are copied from [56].

| Decoder | Fusion (things) | Fusion (stuff) | PQ | AP$^{mask}$ | oIOU |
|---|---|---|---|---|---|
| Unified | | | 45.1 | 42.9 | 67.1 |
| Decoupled | | | 50.6 | 43.6 | 67.6 |
| Unified (Fig. 4a) | ✓ | ✓ | 44.6 | 42.5 | 66.8 |
| Decoupled (Fig. 4b) | ✓ | ✓ | 50.0 | **44.4** | 77.1 |
| Decoupled (Fig. 4c) | ✓ | | **51.3** | **44.4** | **77.3** |

**Table 5:** An ablation study on different decoder and text-image fusion designs, as depicted in Fig. 4. We report PQ for panoptic segmentation on MSCOCO, AP$^{mask}$ for instance segmentation on MSCOCO, and oIoU for referring segmentation on RefCOCO's validation set. Our final choice is highlighted in gray.

its effectiveness and versatility in detecting and segmenting a wide range of object categories in real-world scenarios.

**Part Segmentation.** We evaluate our models performance on Pascal-Panoptic-Parts dataset [5] and report mIoU$_{partS}$ in Table 3. We followed the standard grouping from [5]. Our model outperforms state-of-the-art by +5.2 in this metric. We also provide qualitative comparisons with Grounding DINO + SAM in Fig. 7. Our findings reveal that the results of Grounding SAM are heavily constrained by the detection performance of Grounding DINO. As a result, they are unable to fully leverage the benefits of SAM in producing accurate and fine-grained part segmentation masks.

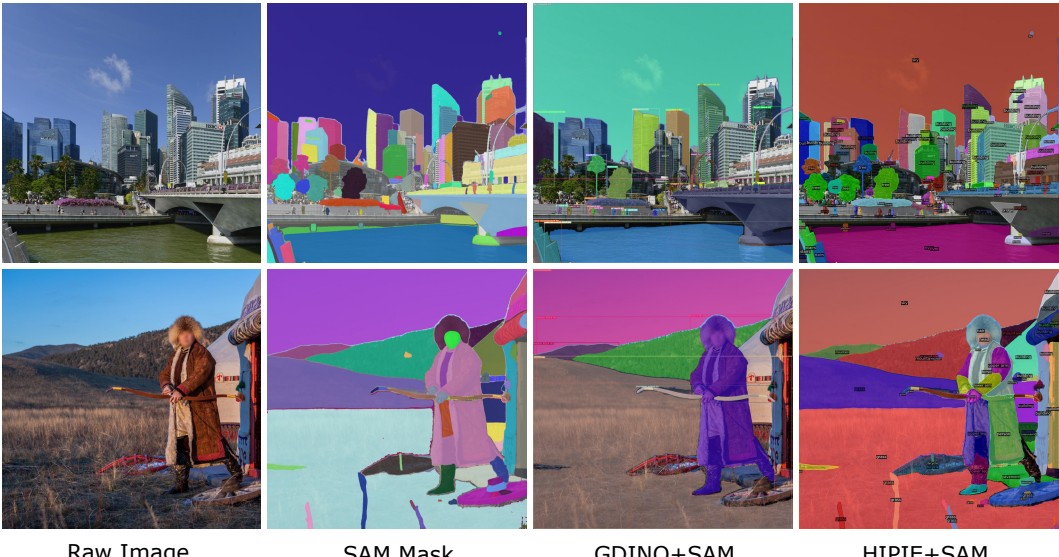

Raw Image    SAM Mask    GDINO+SAM    HIPIE+SAM

**Figure 7:** Results of merging HIPIE with SAM for class-aware image segmentation on SA-1B dataset. Grounded-SAM (Grounding DINO + SAM) [29, 24] cannot fully leverage the benefits of SAM in producing accurate and fine-grained part segmentation masks. Our method demonstrates fewer misclassifications and overlooked masks across the SA-1B dataset compared to the Grounded-SAM approach.

| Method | Backbone | Object Detection | | | |
|---|---|---|---|---|---|
| | | AP | $AP_S$ | $AP_M$ | $AP_L$ |
| Deform. DETR [65] | RN50 | 46.9 | 29.6 | 50.1 | 61.6 |
| DN-DETR [27] | RN50 | 48.6 | 31.0 | 52.0 | 63.7 |
| UNINEXT [58] | RN50 | 51.3 | 32.6 | 55.7 | 66.5 |
| HIPIE | RN50 | 53.9 | 37.5 | 58.0 | 68.0 |
| *vs. prev. SOTA* | | **+2.6** | **+4.9** | **+2.3** | **+1.5** |
| Cas. Mask-RCNN [3] | CNeXtL | 54.8 | - | - | - |
| ViTDet-H [31] | ViT-H | 58.7 | - | - | - |
| UNINEXT [58] | ViT-H | 58.1 | 40.7 | 62.5 | 73.6 |
| HIPIE | ViT-H | 61.3 | 45.8 | 65.7 | 75.9 |
| *vs. prev. SOTA* | | **+3.2** | **+5.1** | **+3.2** | **+2.3** |

**Table 6:** Comparisons on the instance segmentation and object detection tasks. We evaluate model performance on the validation set of MSCOCO.

| Method | Backbone | COCO oIoU | COCO+ oIoU | COCOg oIoU |
|---|---|---|---|---|
| MAttNet [60] | RN101 | 56.5 | 46.7 | 47.6 |
| VLT [9] | Dark56 | 65.7 | 55.5 | 53.0 |
| RefTR [40] | RN101 | 74.3 | 66.8 | 64.7 |
| UNINEXT [58] | RN50 | 77.9 | 66.2 | 70.0 |
| UNINEXT [58] | ViT-H | 82.2 | 72.5 | 74.7 |
| HIPIE | RN50 | 78.3 | 66.2 | 69.8 |
| HIPIE | ViT-H | 82.6 | 73.0 | 75.3 |
| *vs. prev. SOTA* | | **+0.4** | **+0.5** | **+0.6** |

**Table 7:** Comparison on the referring image segmentation (RIS) task. We evaluate the model performance on the validation sets of RefCOCO, RefCOCO+, and RefCOCOg datasets using overall IoU (oIoU) metrics.

**Object Detection and Instance Segmentation.** We evaluate our model's object detection and instance segmentation capabilities following previous works [28, 67, 56]. On MSCOCO [34] and ADE20K [64] datasets, HIPIE achieves an increase of +5.1 and +0.6 AP$^{mask}$ respectively. Detailed comparisons are provided in Sec. 4.2 which demonstrate state-of-the-art results on ResNet and ViT architectures consistently across all Average Precision metrics.

**Referring Segmentation.** Referring image segmentation (RIS) tasks are examined using the Ref-COCO, RefCOCO+, and RefCOCOg datasets. Our model outperforms all the other alternatives by an average of +0.5 in overall IoU (oIoU).

## 4.3 Ablation Study

To demonstrate the effectiveness of our design choices for text-image fusion mechanisms and representation learning modules for stuff and thing classes, we conduct an ablation study (depicted in Fig. 4) and present the results in Table 5. From this study, we draw several important conclusions: *1)* Text-image fusion plays a critical role in achieving accurate referring segmentation results. *2)* The early text-image fusion approach for stuff classes negatively impacts the model's performance on panoptic segmentation. This finding validates our analysis in the introduction section, where we highlighted the challenges introduced by the high levels of confusion in stuff's textual features, which can adversely affect the quality of representation learning. *3)* Our design choices significantly improve the performance of panoptic segmentation, instance segmentation, and referring segmentation tasks. These conclusions underscore the importance of our proposed design choices in achieving improved results across multiple segmentation tasks.

## 5 Conclusions

This paper presents HIPIE, an open-vocabulary, universal, and hierarchical image segmentation model that is capable of performing various detection and segmentation tasks using a unified framework, including object detection, instance-, semantic-, panoptic-, hierarchical-(whole instance, part, subpart), and referring-segmentation tasks. Our key insight is that we should decouple the representation learning modules and text-image fusion mechanisms for background (*i.e.*, referred to as stuff) and foreground (*i.e.*, referred to as things) classes. Extensive experiments demonstrate that HIPIE achieves state-of-the-art performance on diverse datasets, spanning across a wide range of tasks and segmentation granularity.

**Acknowledgement** Trevor Darrell and XuDong Wang were funded by DoD including DARPA LwLL and the Berkeley AI Research (BAIR) Commons.

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

# Appendix

## A.1  List of Datasets

|  | semantic | instance | panoptic | grounding | part | training | # images |
|---|:---:|:---:|:---:|:---:|:---:|:---:|:---:|
| ADE-150 | ✓ | ✓ | ✓ |  |  |  | 2000 |
| Pascal VOC | ✓ |  |  |  |  |  | 1449 |
| Pascal Context-59 | ✓ |  |  |  |  |  | 5105 |
| Pascal-Panoptic-Parts | ✓ | ✓ | ✓ |  | ✓ | * | 10103 |
| COCO | ✓ | ✓ | ✓ |  |  | ✓ | 121408 |
| RefCOCO |  |  |  | ✓ |  | ✓ | 19994 |
| RefCOCO+ |  |  |  | ✓ |  | ✓ | 19992 |
| RefCOCOg |  |  |  | ✓ |  | ✓ | 26711 |

**Table A1: List of the dataset used.** The checkmarks denote whether a dataset has a particular type of annotation and whether the dataset is used in the training process. * Because of a data leak between Pascal-Panoptic-Parts and other Pascal datasets, we use weights trained without Pascal-Panoptic-Parts in those evaluations unless otherwise specified.

We report the statistics of datasets used in training and evaluation in table Table A1. Additionally, we further evaluate our model on 35 object detection datasets and 25 segmentation datasets in Sec. A.4.2. In total, we benchmarked our model on around 70 datasets. These benchmarks show our model can adapt to many different scenarios and retain a reasonable performance in a zero-shot manner.

## A.2  Hierarchical Segmentation

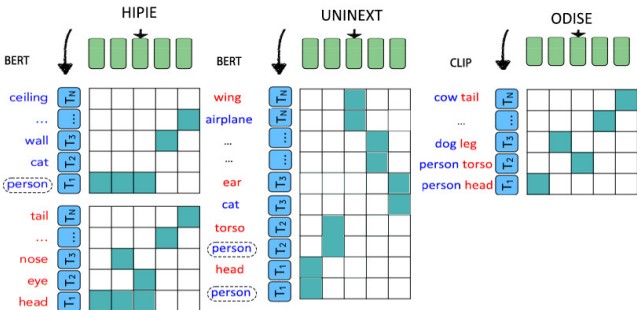

**Figure A1: Hierarchal Design of HIPIE compared with other methods**.

Fig. A1 highlights the difference of our approach with other methods for hierarchical segmentation. We concatenate class names of different hierarchies as prompts. During the training, we uniquely contrast a mask embedding with both scene-level and part-level labels explicitly. Previous works such as UNINEXT and ODISE only treat these classes as normal multi-word labels. While UNINEXT allows contrasting different words individually because of the design of BERT encoder, it leads to suboptimal signals. In the example above, "person head" has both positive and negative target for "person".

## A.3  Experiment Setup

### A.3.1  Model Learning Settings

HIPIE is first pre-trained on Objects365 [48] for 340k iterations, using a batch size of 64 and a learning rate of 0.0002, and the learning rate is dropped by a factor of 10 after the 90th percentile of the schedule. After the pre-training stage, we finetune HIPIE on COCO [34], RefCOCO, RefCOCOg, and RefCOCO+ [41, 61] jointly for 120k iterations, using a batch size of 32 and a learning rate of 0.0002. For both stages, we resize the original images so that the shortest side is at least 800

pixels and at most 1024 pixels, while the longest side is at most 1333 pixels. For part segmentation, we train additionally train our model jointly on Pascal-Panoptic-Parts [5] dataset and all previously mentioned datasets. Because of potential data leaks between Pascal-Panoptic-Parts and other Pascal datasets used in the open-vocabulary segmentation evaluation, we report those numbers with weights not trained on Pascal-Panoptic-Part dataset. Because of our hierarchal design, our model produces better-quality masks. In particular, our model can generalize to novel hierarchies that do not exist in part segmentation datasets. In Fig. 6, we provide visualization of such results.

### A.3.2 Implementation Details

For loss functions in Eq. (3), we have $\lambda_{cls} = 2.0, \lambda_{mask} = 5.0, \lambda_{box} = 5.0, \lambda_{ce} = 1.0, \lambda_{dice} = 1.0, \lambda_{L1} = 1.0, \lambda_{giou} = 0.2$. For $\lambda$ in Eq. (4), we use $\lambda = 0.2$ for seen classes during the training and $\lambda = 0.45$ for novel classes. In close-set evaluation, we set $\lambda = 0.0$ and do not use CLIP. We also do not use CLIP for PAS-21 evaluation (whose classes are mostly covered by COCO) because we find it degrades the performance. We use 800 and 1024-resolution images during the training. For evaluations, we use 1024-resolution images.

### A.3.3 Training Process

| Stage | Task | Dataset | Batch Size | Max Iter | Step |
|-------|------|---------|-----------|----------|------|
| I | OD&IS | Objects365 | 64 | 340741 | 312346 |
| II | OD&IS
REC&RIS | COCO
RefCOCO/g/+ | 32
32 | 91990 | 76658 |
| III | PanoS
REC&RIS
PartS | COCO
RefCOCO/g/+
Pascal-Panoptic-Parts | 32
32
32 | 150000 | 100000,135000 |

**Table A2: Training Process.** Following UNINEXT [58], We first pretrain our model for object detection on Object365 for 340k iteration (Stage I). Then we fine-tune our model jointly on COCO for object detection, instance segmentation, referring expression comprehension (REC), and referring segmentation (RIS) for 92k iteration (Stage II). We further jointly train our model on Panoptic Segmentation, REC, RIS, and Part Segmentation for 150k iteration (Stage III)

We train all our models on NVIDIA-A100 GPUs with a batch size of 2 per GPU using AdamW [38] optimizer. We use a base learning rate of 0.0001 and a weight decay of 0.01. The learning rate of the backbone is further multiplied by 0.1. Following UNINEXT [58], We first pretrain our model for object detection on Object365 for 340k iteration (Stage I). Then we fine-tune our model jointly on COCO for object detection, instance segmentation, referring expression comprehension (REC), and referring segmentation (RIS) for 91k iteration (Stage II). We further jointly train our model on Panoptic Segmentation, REC, RIS, and Part Segmentation for 150k iteration (Stage III). In Stage I, the learning rate is dropped by a factor of 10 after 312k iterations. In stage II, the learning rate is dropped by a factor of 10 after 77k iterations. In Stage III, the learning rate is dropped by a factor of 10 after 100k and 135k iterations. In all stages, we sample uniformly across datasets when there are multiple datasets. The global batch size is 64 in Stage I and 32 in Stage II and III. Notably, our stage I and II is identical to the setup of UNINEXT. For ablation studies, we train stage III only and reduce the schedule to 90k iterations. The learning rate schedule is also scaled accordingly. The details of training recipe is shown in Table A2.

## A.4 Additional Evaluations

### A.4.1 Referring Expression Comprehension

In addition to Referring Segmentation reported in Table 7, we further report results on Referring Expression Comprehension (REC), which aims to locate a target object in an image at the pixel-level, given a referring expression as input. We establish new state-of-the-art performance by an average of $+0.3$ P@0.5 and $+0.5$ oIoU across three datasets.

| Method | Backbone | RefCOCO | | RefCOCO+ | | RefCOCOg | |
|---|---|---|---|---|---|---|---|
| | | oIoU | P@0.5 | oIoU | P@0.5 | oIoU | P@0.5 |
| MAttNet [60] | RN101 | 56.5 | 76.7 | 46.7 | 65.3 | 47.6 | 66.6 |
| VLT [9] | Dark56 | 65.7 | 76.2 | 55.5 | 64.2 | 53.0 | 61.0 |
| RefTR [40] | RN101 | 74.3 | 85.7 | 66.8 | 77.6 | 64.7 | 82.7 |
| UNINEXT [58] | RN50 | 77.9 | 89.7 | 66.2 | 79.7 | 70.0 | 84.0 |
| UNINEXT [58] | ViT-H | 82.2 | 92.6 | 72.5 | 85.2 | 74.7 | 88.7 |
| HIPIE | RN50 | 78.3 | 90.1 | 66.2 | 80.0 | 69.8 | 83.6 |
| HIPIE | ViT-H | 82.6 | 93.0 | 73.0 | 85.5 | 75.3 | 88.9 |
| *vs. prev. SOTA* | | **+0.4** | **+0.4** | **+0.5** | **+0.3** | **+0.6** | **+0.2** |

**Table A3:** Comparison on the referring expression comprehension (REC), and referring image segmentation (RIS) tasks. The evaluation is carried out on the validation sets of RefCOCO, RefCOCO+, and RefCOCOg datasets using Precision@0.5 and overall IoU (oIoU) metrics for REC and RIS, respectively.

| Method | Mean | Median | AerialMaritime Drone large | AerialMaritime Drone tiled | American Sign Lang Letters | Aquarium | BCCD | boggleBoards | brackish Underwater | ChessPieces | Cottontail Rabbits | dice | mediumColor | DroneControl | EgoHands generic | EgoHands specific | HardHat Workers | Mask Wearing | Mountain Dew Commercial | North America Mushrooms | openPoetry Vision | Oxford Pets by-breed | Oxford Pets by-species | Packages | Pascal VOC | Pistols | PKLot | plantdoc | Pothole | Raccoon | selfdriving Car | Shellfish OpenImages | Thermal Cheetah | thermal Dogs And People | Uno Cards | Vehicles OpenImages | website Screenshots | Wildfire Smoke |
|---|---|---|---|---|---|---|---|---|---|---|---|---|---|---|---|---|---|---|---|---|---|---|---|---|---|---|---|---|---|---|---|---|---|---|---|---|---|---|
| MDETR | 10.7 | 3.0 | 0.6 | 5.4 | 0.3 | 1.7 | 6.7 | 0.0 | 0.7 | 3.0 | 66.5 | 0.0 | 3.8 | 5.9 | **3.5** | 0.4 | 0.4 | 3.0 | 39.8 | 0.0 | 0.0 | 0.7 | 63.6 | 5.6 | 15.9 | 0.0 | 0.5 | **12.7** | **50.6** | 2.8 | 8.1 | **4.5** | 42.8 | 0.0 | 13.4 | **0.7** | 12.5 |
| GLIP-T | 11.4 | 1.6 | 8.3 | **17.1** | 0.1 | 16.0 | 1.7 | 0.0 | 1.7 | 0.0 | 57.0 | 0.5 | 0.1 | 1.1 | 0.1 | **2.7** | 0.6 | 15.3 | 5.9 | 0.0 | 0.3 | 1.6 | 58.3 | 51.2 | 31.6 | 0.0 | 1.6 | 6.2 | **7.4** | 15.9 | 0.2 | 38.7 | 0.0 | **55.0** | 0.3 | 0.0 |
| HIPIE † | 14.5 | 3.9 | 5.2 | 9.6 | **2.9** | 8.6 | 6.0 | 0.0 | 0.9 | 3.8 | 69.5 | **0.5** | 0.7 | 5.8 | 0.2 | 1.4 | 0.8 | **37.7** | 27.4 | 0.0 | **7.8** | 2.5 | **68.1** | 58.6 | 36.4 | 1.1 | **3.7** | 3.9 | 33.4 | 5.3 | 27.5 | 0.5 | 24.5 | 0.0 | 53.9 | 0.3 | 0.0 |
| HIPIE ‡ | **17.9** | **5.5** | **10.9** | 16.6 | 2.8 | **18.3** | **8.0** | 0.1 | **2.7** | **5.5** | **75.7** | 0.3 | **1.6** | **6.6** | 0.5 | 1.8 | **1.1** | 8.5 | **42.7** | 0.0 | 7.2 | **2.7** | 56.2 | **66.0** | **66.8** | **2.6** | 3.6 | 2.9 | 49.7 | **7.3** | **49.6** | 0.3 | **53.3** | 0.0 | 53.6 | 0.4 | 0.3 |

**Table A4:** We present the object detection results in the OdinW benchmark. We report mAP and mean results averaged over 35 datasets. Notably, our ResNet-50 baseline surpasses GLIP-T by +3.1. We use the notation HIPIE † and HIPIE ‡ to denote our method with ResNet-50 and ViT-H backbones, respectively.

| Method | Mean | Median | Airplane Parts | Bottles | Brain Tumor | Chicken | Cows | Electric Shaver | Elephants | Fruits | Garbage | Ginger Garlic | Hand | Hand Metal | House Parts | HouseHold Items | Nutterfly Squireel | Phones | Poles | Puppies | Rail | Salmon Fillet | Strawberry | Tablets | Toolkits | Trash | Watermelon |
|---|---|---|---|---|---|---|---|---|---|---|---|---|---|---|---|---|---|---|---|---|---|---|---|---|---|---|---|
| X-Decoder(L) | 32.3 | 22.3 | 13.1 | 42.1 | **2.2** | 8.6 | 44.9 | 7.5 | 66.0 | **79.2** | **33.0** | 11.6 | 75.9 | 42.1 | **7.0** | 53.0 | 68.4 | 15.6 | 20.1 | 59.0 | **2.3** | 19.0 | 67.1 | **22.5** | 9.9 | 22.3 | 13.8 |
| HIPIE (H) | **41.2** | **45.1** | **14.0** | **45.1** | 1.9 | **46.5** | **50.1** | **76.1** | **68.6** | 61.1 | 31.2 | **24.3** | **94.2** | **64.0** | 6.8 | **53.4** | **79.7** | 7.0 | 6.7 | **64.6** | 2.2 | **41.8** | **81.5** | 8.8 | **17.9** | **31.2** | **50.6** |

**Table A5:** Segmentation Result on SeginW benchmark across 25 datasets. We report mAP. We outperform X-Decoder by a large margin (+8.9)

#### A.4.2 Object Detection and Segmentation in the Wild

To further examine the open-vocabulary capability of our model, we evaluate it on the Segmentation in the Wild (SeginW) [66] consisting of 25 diverse segmentation datasets and Object Detection in the Wild (OdinW) [29] Benchmark consisting of 35 diverse detection datasets. Since OdinW benchmark contains Pascal VOC and some of the classes in SeginW benchmark are covered by Pascal-Panoptic-Parts, we use a version of our model that is not trained on Pascal-Panoptic-Parts for both benchmarks for a fair comparison.

We report the results in Table A5 and Table A4. Notably, our method establishes a new state-of-the-art of SeginW benchmark by an average of +8.9 mAP across 25 datasets. We achieve comparable performance under similar settings. In particular, our ResNet-50 baseline outperforms GLIP-T by +3.1 mAP. We note that other methods such as GroundingDINO [36] achieve better absolute performance by introducing more grounding data, which can be critical in datasets whose classes are not common objects. (For example, the classes of Boggle Boards are letters, the classes of UnoCards are numbers, and the classes of websiteScreenshots are UI elements).

### A.5 Other Ablation Studies

We provide further ablations on a few design choices in this section.

**Text Encoder**. We experiment with replacing the BERT text encoder in UNINEXT with a pre-trained CLIP encoder. Additionally, following practices of ODISE [56], we prompt each label to a sentence "a photo of <label>". For RIS and REC tasks, the language expression remains unchanged. We report

|  | COCO | | RefCOCO |
|  | PQ | AP$^{Mask}$ | oIoU |
|---|---|---|---|
| CLIP | **51.5** | 44.3 | 48.7 |
| BERT | 51.3 | **44.4** | **77.3** |

**Table A6:** Ablation Studies on the choice of Text Encoder. We find that while CLIP and BERT achieve similar performance on panoptic and instance segmentation, BERT performs significantly better on Referring Instance Segmentation (+28.6 oIoU).

|  | COCO | | RefCOCO |
|  | PQ | AP$^{Mask}$ | oIoU |
|---|---|---|---|
| w/o OTA | 50.9 | 43.6 | 76.3 |
| w/ OTA | **51.3** | **44.4** | **77.3** |

**Table A7:** Ablation Studies on the SimOTA matching process. Introducing SimOTA leads to performance improvement in all evaluation metrics.

the result in Table A6. We find that while CLIP and BERT achieve similar performance on panoptic and instance segmentation, BERT performs significantly better on referring instance segmentation (+28.6 oIoU). We hypothesize that this may be caused by the lack of explicit language-focused training which can help achieve a better understanding of referring expression.

**SimOTA**.Following UNINEXT [58] we adopted simOTA in the matching process for "thing" classes during the training. We experiment with removing simOTA matching and use standard one-to-one matching instead. We report the result in Table A7. We find that simOTA improves the performance on both panoptic segmentation and referring instance segmentation.

## A.6 Limitations

We've showcased experimental evidence supporting our method across a diverse set of tasks, including open vocabulary panoptic and semantic segmentation, instance and referring segmentation, and object detection. However, it will be crucial for future work to test our methodology on video-related tasks, such as object tracking and segmentation, to draw comparisons with other universal models like UNINEXT [58]. Furthermore, it's worth considering additional pretraining of our vision encoder on newer, more complex datasets that encompass a vast amount of masks and information. For instance, SA-1B [24], which includes over 1 billion masks, would serve as an ideal training ground. Lastly, it would be advantageous to measure the change in performance when training on supplementary hierarchical datasets. Such an approach will allow us to demonstrate more varied object part segmentations, thereby expanding the capabilities and versatility of our model.

## A.7 Broader Impact

Our research introduces a potent approach to hierarchical and universal open vocabulary image segmentation, aiming to address the ever-increasing demand for more data and advanced model architectures. As the demand increases, practical methodologies such as universal segmentation are projected to play a vital role in constructing and training foundational models. Our model, HIPIE, shows promise for fostering progress in a multitude of fields where hierarchical data are critical, including self-driving cars, manufacturing, and medicine. However, it's imperative to acknowledge potential limitations. Given that our model is trained on human annotations and feedback, it can inadvertently replicate any errors or biases present in the datasets. The architecture's complexity is further enhanced when multiple models are integrated, which, in turn, compromises the explainability of the final predictions. Therefore, as with the introduction of any novel technology, it's crucial to implement safety protocols to mitigate misuse. This includes mechanisms for ensuring the accuracy of inputs and establishing procedures to comprehend the criteria the model employs for predictions. By doing so, we can improve the model's reliability and mitigate potential issues.

## A.8 Qualitative Results

### A.8.1 More Visualizations

We provide more visualizations of panoptic segmentation, part segmentation and referring segmentation in Figs. A2 and A3.

### A.8.2 Combining with SAM

We integrate our model with the mask outputs generated by the ViT-H Image encoder from Segment Anything (SAM) [24]. The encoder is trained on SA-1B which encompasses a broad spectrum of objects and masks within each image, enabling us to enhance our segmentation output by utilizing the high-quality masks from the SAM encoder to generate finer, more detailed masks.

To elaborate, in the context of panoptic segmentation, we implement a voting scheme between our pixel-wise annotations and the masks from Segment Anything (SAM), enriching these masks with our labels. For objects where our model demonstrates a strong understanding of hierarchy, such as "person" or "bird", we substitute the SAM masks with ours. This approach enables us to optimize hierarchical outcomes in the face of highly complex images.

Based on our observations from the figures, it's evident that Grounding DINO generates instance segmentation bounding boxes and subsequently uses SAM for the application of the segmentation masks. While this method proves effective for most datasets, SA-1B is a highly complex set featuring a vast array of whole objects, parts and subparts. Our qualitative findings suggest that the a single granularity instance segmentation model may fail to fully capture all objects/parts within an image or may incorrectly identify them. This consequently leads to SAM receiving sub-optimal bounding boxes for segmentation, resulting in fewer and less accurate masks (see third columns in Figs. A4 to A6). In contrast, our methodology (see last columns in Figs. A4 to A6) integrates the SAM encoder masks with our annotations and hierarchical masks wherever feasible. This results in a significantly more fine-grained and accurate output, proving superior in handling complex datasets such as SA-1B.

### A.8.3 Combining with Stable Diffusion

As an interesting experiment, we combined our model with image generation model Stable-Diffusion[46] in Fig. A7. Given a source expression and target prompt, we first use HIPIE's segmentation capability to find the corresponding masks, which are then used for image inpainting. Notably, our model can uniquely achieve fine-grained control over object parts by providing part segmentation masks.

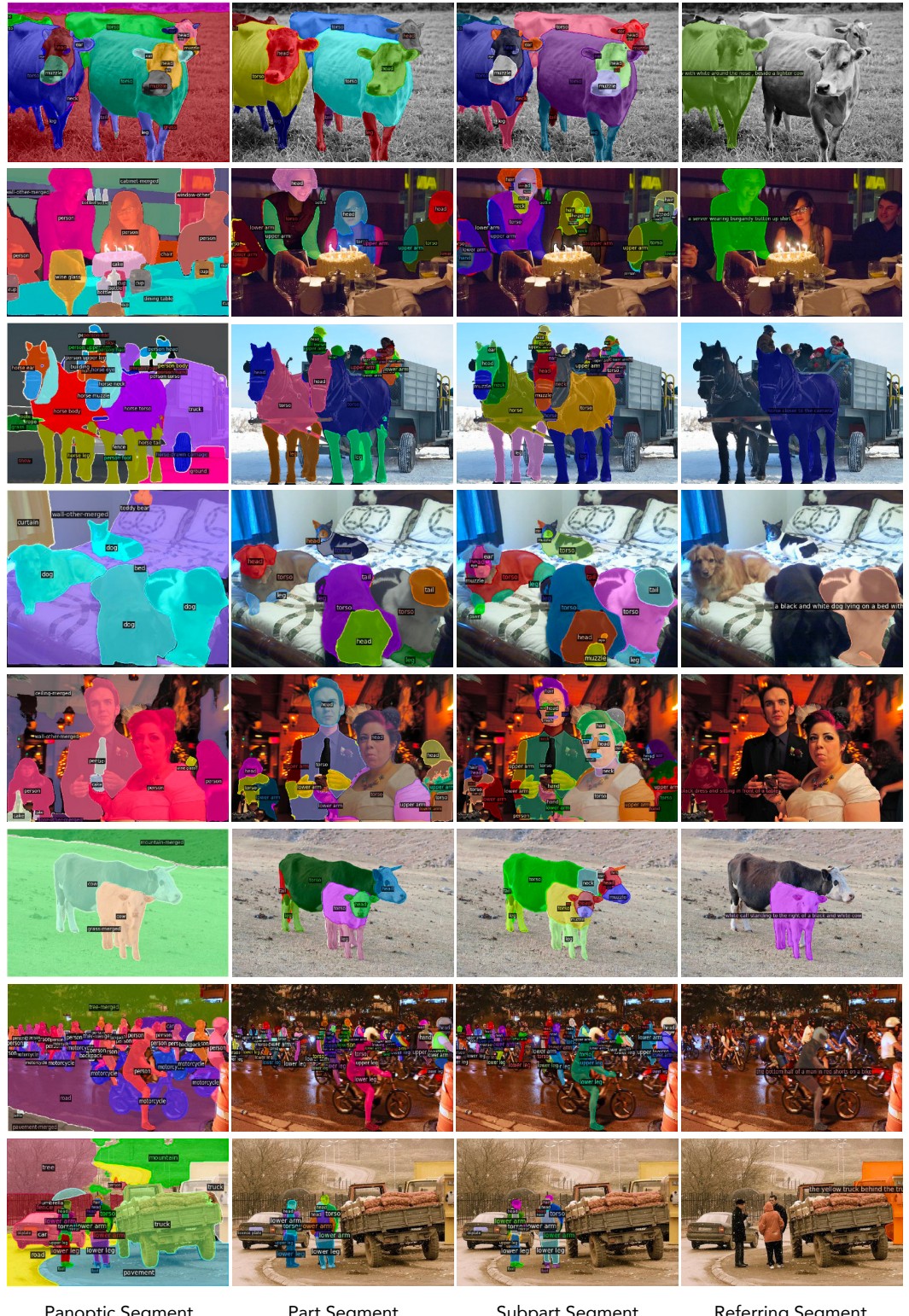

| Panoptic Segment | Part Segment | Subpart Segment | Referring Segment |

**Figure A2:** More visualizations showcasing panoptic segmentation, part segmentation, subpart segmentation, and referring segmentation results on RefCOCO. It is recommended to view the results in color and zoom in for better detail.

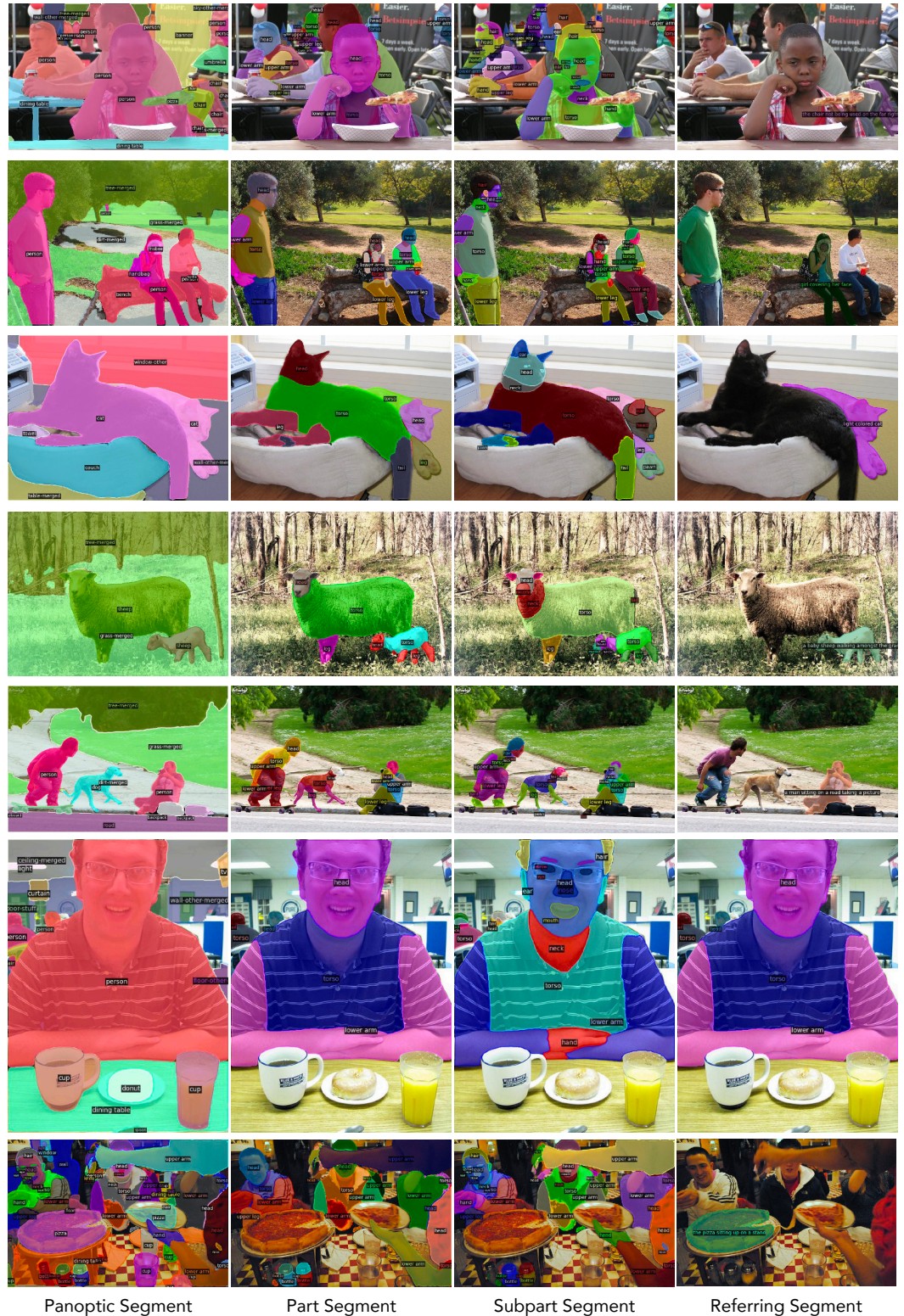

Panoptic Segment      Part Segment      Subpart Segment      Referring Segment

**Figure A3:** More visualizations showcasing panoptic segmentation, part segmentation, subpart segmentation, and referring segmentation results on RefCOCO. It is recommended to view the results in color and zoom in for better detail.

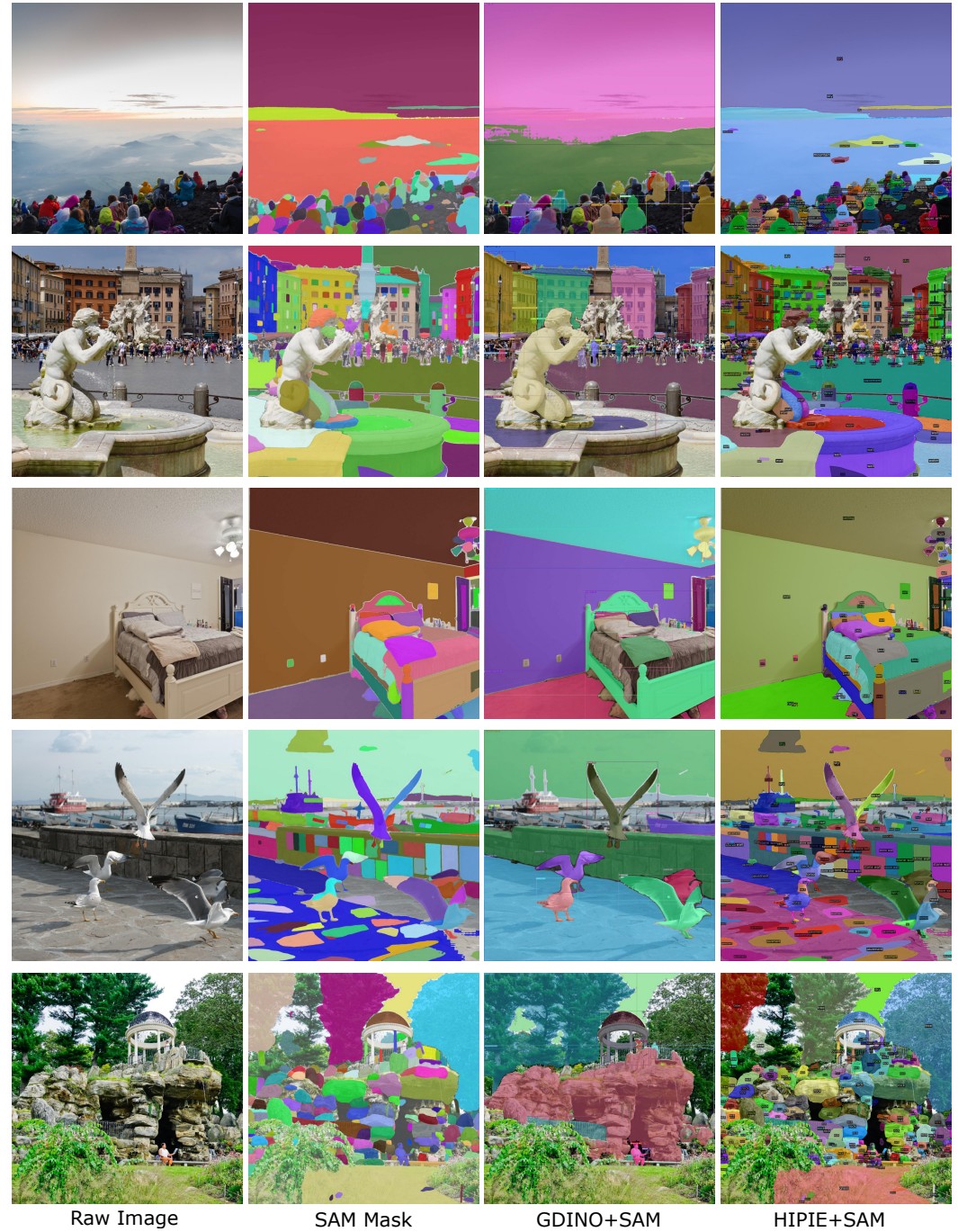

| Raw Image | SAM Mask | GDINO+SAM | HIPIE+SAM |

**Figure A4:** Additional results of merging HIPIE with SAM for hierarchical segmentation. By integrating the part masks from our model and conducting a vote among SAM's panoptic masks, we generate finely detailed mask outputs. Our method demonstrates fewer misclassifications and overlooked masks across the SA-1B dataset compared to the Grounding DINO + SAM approach. Furthermore, our technique excels in differentiating between intra-class objects and identifying distinct object parts.

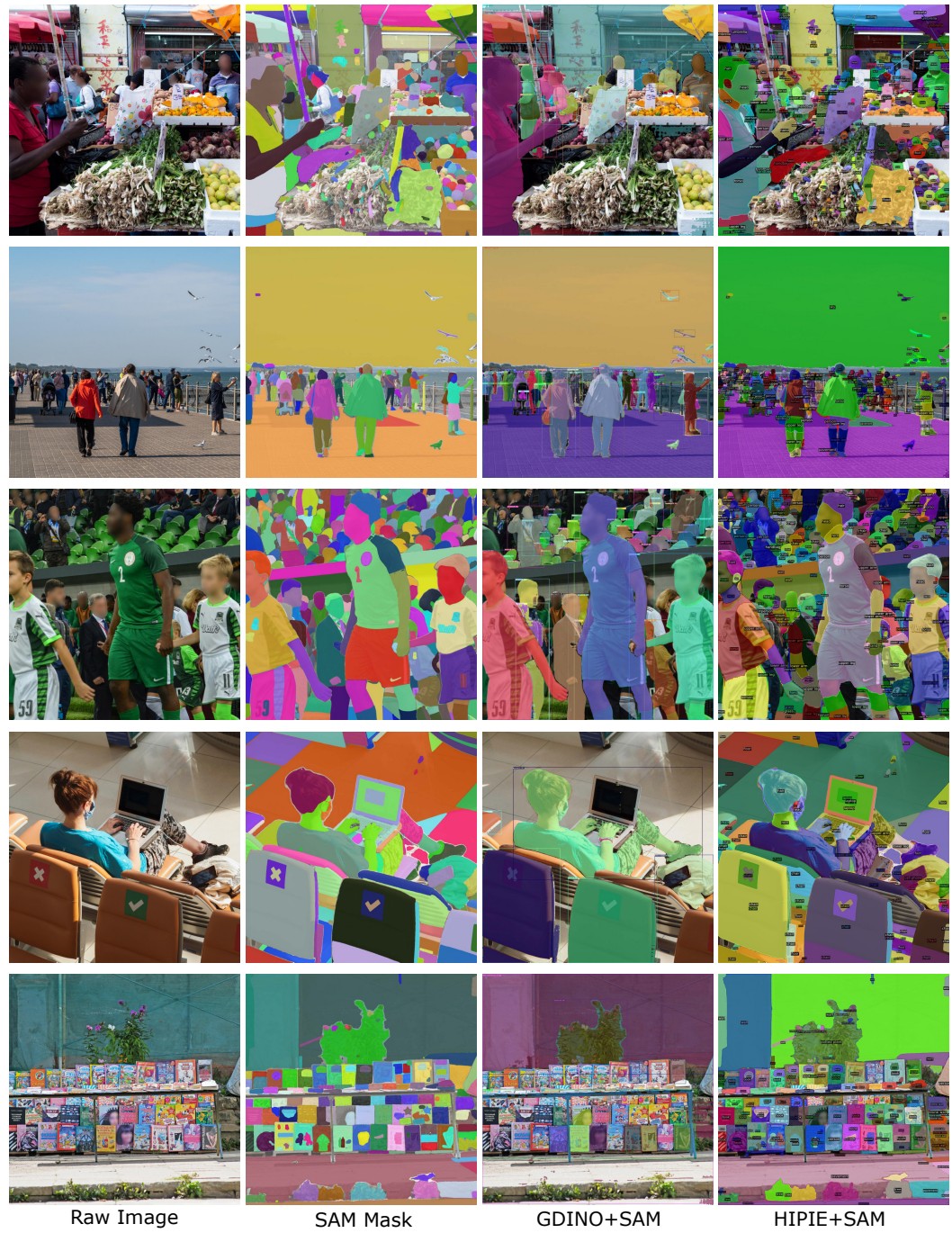

| Raw Image | SAM Mask | GDINO+SAM | HIPIE+SAM |
|---|---|---|---|

**Figure A5:** Additional results of merging HIPIE with SAM for hierarchical segmentation. By integrating the part masks from our model and conducting a vote among SAM's panoptic masks, we generate finely detailed mask outputs. Our method demonstrates fewer misclassifications and overlooked masks across the SA-1B dataset compared to the Grounding DINO + SAM approach. Furthermore, our technique excels in differentiating between intra-class objects and identifying distinct object parts.

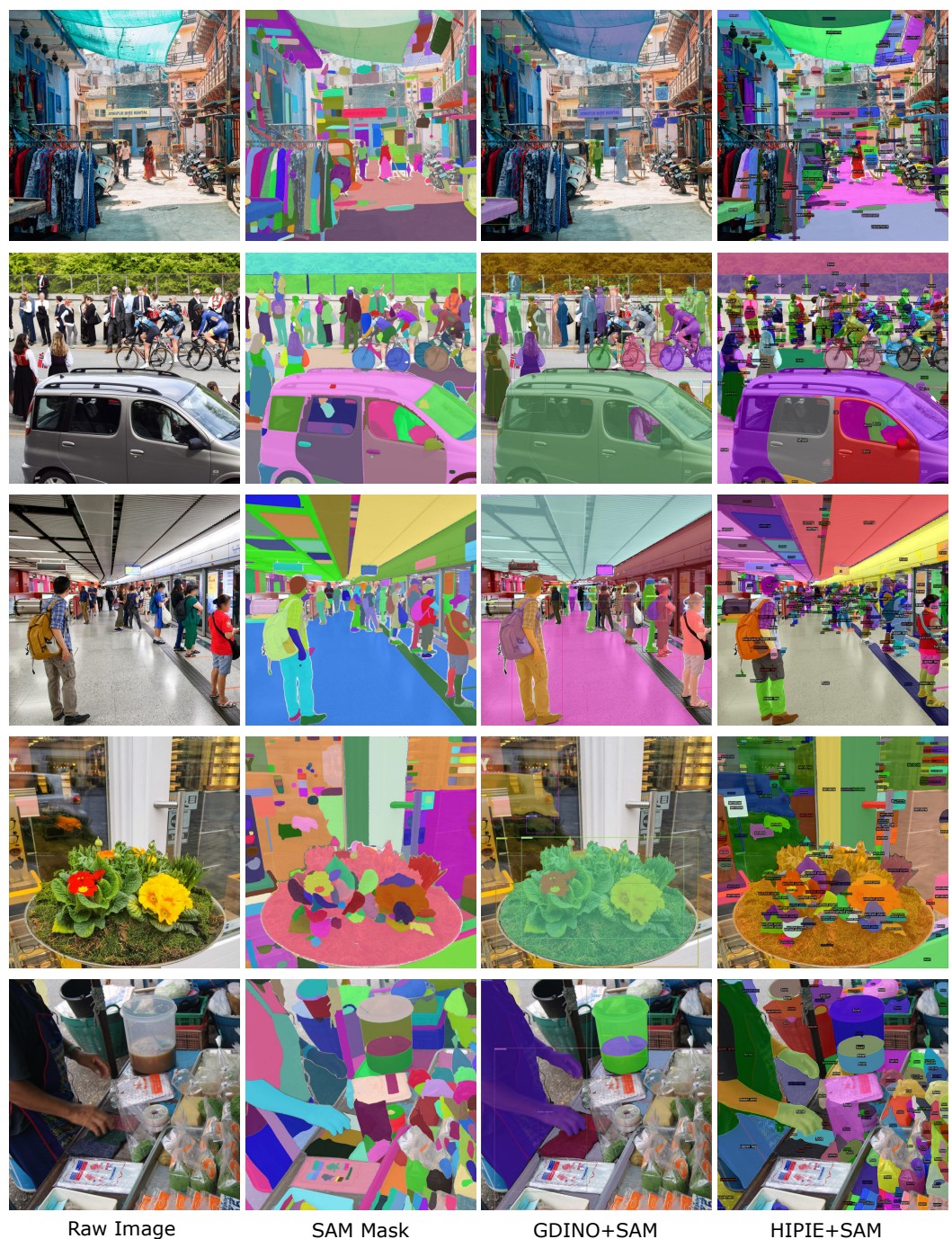

| Raw Image | SAM Mask | GDINO+SAM | HIPIE+SAM |

**Figure A6:** Additional results of merging HIPIE with SAM for hierarchical segmentation. By integrating the part masks from our model and conducting a vote among SAM's panoptic masks, we generate finely detailed mask outputs. Our method demonstrates fewer misclassifications and overlooked masks across the SA-1B dataset compared to the Grounding DINO + SAM approach. Furthermore, our technique excels in differentiating between intra-class objects and identifying distinct object parts.

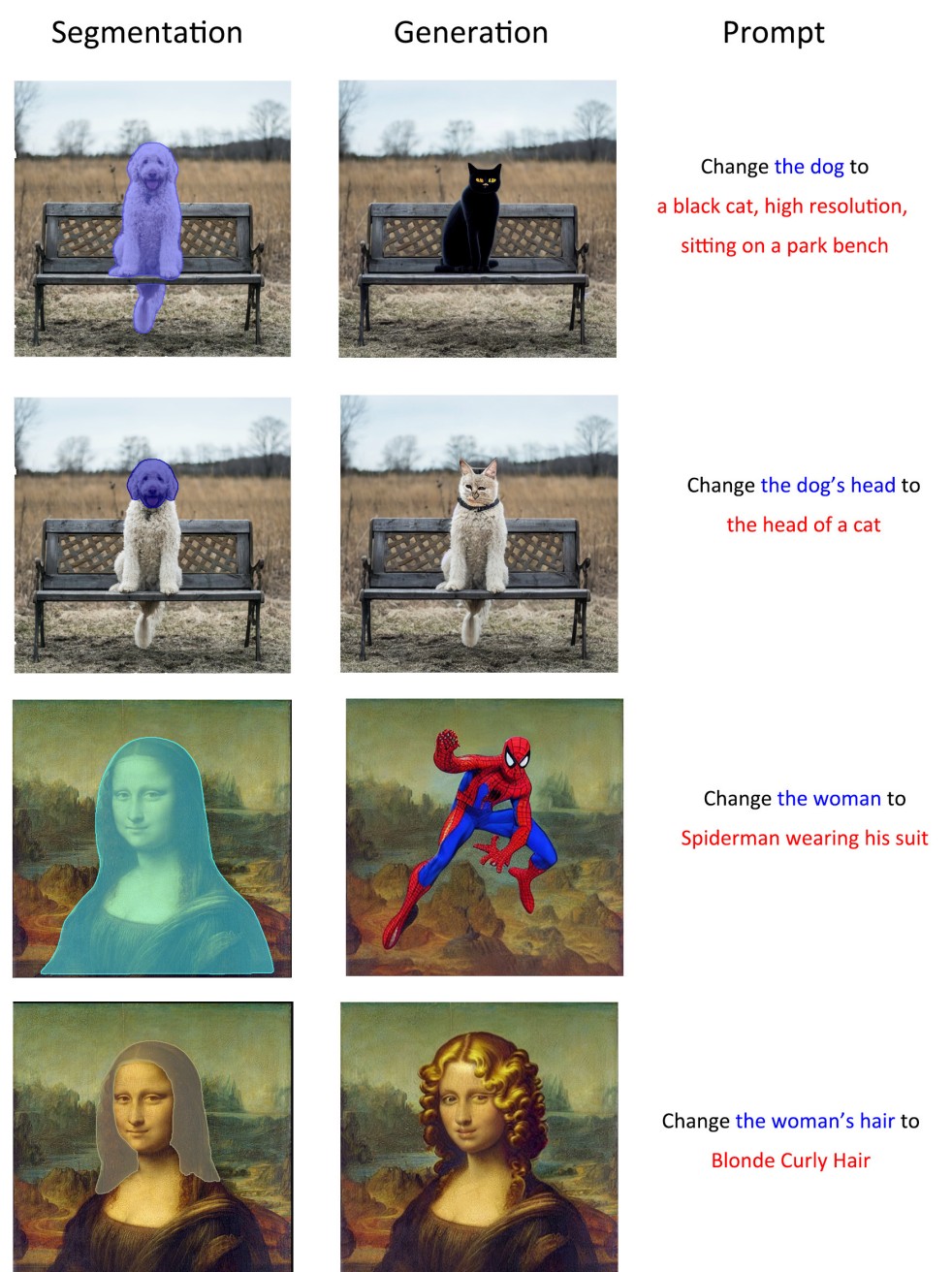

Segmentation        Generation        Prompt

Change the dog to a black cat, high resolution, sitting on a park bench

Change the dog's head to the head of a cat

Change the woman to Spiderman wearing his suit

Change the woman's hair to Blonde Curly Hair

**Figure A7:** Results of combining HIPIE with Stable Diffusion for Image inpainting. We leverage our segmentation model to generate masks for the redrawing process. Our model can uniquely achieve fine-grained control by providing part segmentation masks.

