# Appendix

## A.1 List of Datasets

| | semantic | instance | panoptic | grounding | part | training | # images |
|---|---|---|---|---|---|---|---|
| ADE-150 | ✓ | ✓ | ✓ | | | | 2000 |
| Pascal VOC | ✓ | | | | | | 1449 |
| Pascal Context-59 | ✓ | | | | | | 5105 |
| Pascal-Panoptic-Parts | ✓ | ✓ | ✓ | | ✓ | * | 10103 |
| COCO | ✓ | ✓ | ✓ | | | ✓ | 121408 |
| RefCOCO | | | | ✓ | | ✓ | 19994 |
| RefCOCO+ | | | | ✓ | | ✓ | 19992 |
| RefCOCOg | | | | ✓ | | ✓ | 26711 |

**Table A1: List of the dataset used.** The checkmarks denote whether a dataset has a particular type of annotation and whether the dataset is used in the training process. * Because of a data leak between Pascal-Panoptic-Parts and other Pascal datasets, we use weights trained without Pascal-Panoptic-Parts in those evaluations unless otherwise specified.

We report the statistics of datasets used in training and evaluation in table Table A1. Additionally, we further evaluate our model on 35 object detection datasets and 25 segmentation datasets in Sec. A.3.2. In total, we benchmarked our model on around 70 datasets. These benchmarks show our model can adapt to many different scenarios and retain a reasonable performance in a zero-shot manner.

## A.2 Experiment Setup

### A.2.1 Implementation Details

For loss functions in Eq. (3), we have $\lambda_{cls} = 2.0, \lambda_{mask} = 5.0, \lambda_{box} = 5.0, \lambda_{ce} = 1.0, \lambda_{dice} = 1.0, \lambda_{L1} = 1.0, \lambda_{giou} = 0.2$. For $\lambda$ in Eq. (4), we use $\lambda = 0.2$ for seen classes during the training and $\lambda = 0.45$ for novel classes. In close-set evaluation, we set $\lambda = 0.0$ and do not use CLIP. We also do not use CLIP for PAS-21 evaluation (whose classes are mostly covered by COCO) because we find it degrades the performance. We use 800 and 1024-resolution images during the training. For evaluations, we use 1024-resolution images.

### A.2.2 Training Process

| Stage | Task | Dataset | Batch Size | Max Iter | Step |
|---|---|---|---|---|---|
| I | OD&IS | Objects365 | 64 | 340741 | 312346 |
| II | OD&IS | COCO | 32 | 91990 | 76658 |
| | REC&RIS | RefCOCO/g/+ | 32 | | |
| III | PanoS | COCO | 32 | 150000 | 100000,135000 |
| | REC&RIS | RefCOCO/g/+ | 32 | | |
| | PartS | Pascal-Panoptic-Parts | 32 | | |

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

| | HIPIE | | GLIP-T [32] | MDETR[24] |
|---|---|---|---|---|
| | ViT-H | R50 | Swin-T | EffNet-B5 |
| Pretraining Data | O365,COCO,RefCOCO | | O365 | GOLDG,RefCOCO |
| Mean | **17.9** | 14.5 | 11.4 | 10.7 |
| Median | **5.5** | 3.9 | 1.6 | 3.0 |
| AerialMaritimeDrone_large | **10.9** | 5.2 | 8.3 | 0.6 |
| AerialMaritimeDrone_tiled | 16.6 | 9.6 | **17.1** | 5.4 |
| AmericanSignLanguageLetters | 2.8 | **2.9** | 0.1 | 0.3 |
| Aquarium | **18.3** | 8.6 | 16.0 | 1.7 |
| BCCD | **8.0** | 6.0 | 1.7 | 6.7 |
| boggleBoards | **0.1** | 0.0 | 0.0 | 0.0 |
| brackishUnderwater | **2.7** | 0.9 | 1.7 | 0.7 |
| ChessPieces | **5.5** | 3.8 | 0.0 | 3.0 |
| CottontailRabbits | **75.7** | 69.5 | 57.0 | 66.5 |
| dice_mediumColor | 0.3 | **0.5** | 0.5 | 0.0 |
| DroneControl | **1.6** | 0.7 | 0.1 | 3.8 |
| EgoHands_generic | **6.6** | 5.8 | 1.1 | 5.9 |
| EgoHands_specific | 0.5 | 0.2 | 0.1 | **3.5** |
| HardHatWorkers | 1.8 | 1.4 | **2.7** | 0.4 |
| MaskWearing | **1.1** | 0.8 | 0.6 | 0.4 |
| MountainDewCommercial | 8.5 | **37.7** | 15.3 | 3.0 |
| NorthAmericaMushrooms | **42.7** | 27.4 | 5.9 | 39.8 |
| openPoetryVision | 0.0 | 0.0 | 0.0 | 0.0 |
| OxfordPets_by-breed | 7.2 | **7.8** | 0.3 | 0.0 |
| OxfordPets_by-species | **2.7** | 2.5 | 1.6 | 0.7 |
| Packages | 56.2 | **68.1** | 58.3 | 63.6 |
| Pascal VOC | **66.0** | 58.6 | 51.2 | 5.6 |
| Pistols | **66.8** | 36.4 | 31.6 | 15.9 |
| PKLot | **2.6** | 1.1 | 0.0 | 0.0 |
| plantdoc | 3.6 | **3.7** | 1.6 | 0.5 |
| Pothole | 2.9 | 3.9 | 1.6 | **12.7** |
| Raccoon | 49.7 | 33.4 | 6.2 | **50.6** |
| selfdrivingCar_fixedLarge_export_ | 7.3 | 5.3 | **7.4** | 2.8 |
| ShellfishOpenImages | **49.6** | 27.5 | 15.9 | 8.1 |
| ThermalCheetah | 0.3 | 0.5 | 0.2 | **4.5** |
| thermalDogsAndPeople | **53.3** | 24.5 | 38.7 | 42.8 |
| UnoCards | 0.0 | 0.0 | 0.0 | 0.0 |
| Vehicles-OpenImages | 53.5 | 53.9 | **55.0** | 13.4 |
| websiteScreenshots | 0.4 | 0.3 | 0.3 | **0.7** |
| WildfireSmoke | 0.3 | 0.0 | 0.0 | 12.5 |

**Table A7:** Object Detection Result in OdinW benchmark. We report mAP. We achieve comparable performance under similar settings. In particular, our ResNet-50 baseline outperforms GLIP-T by $+3.1$.

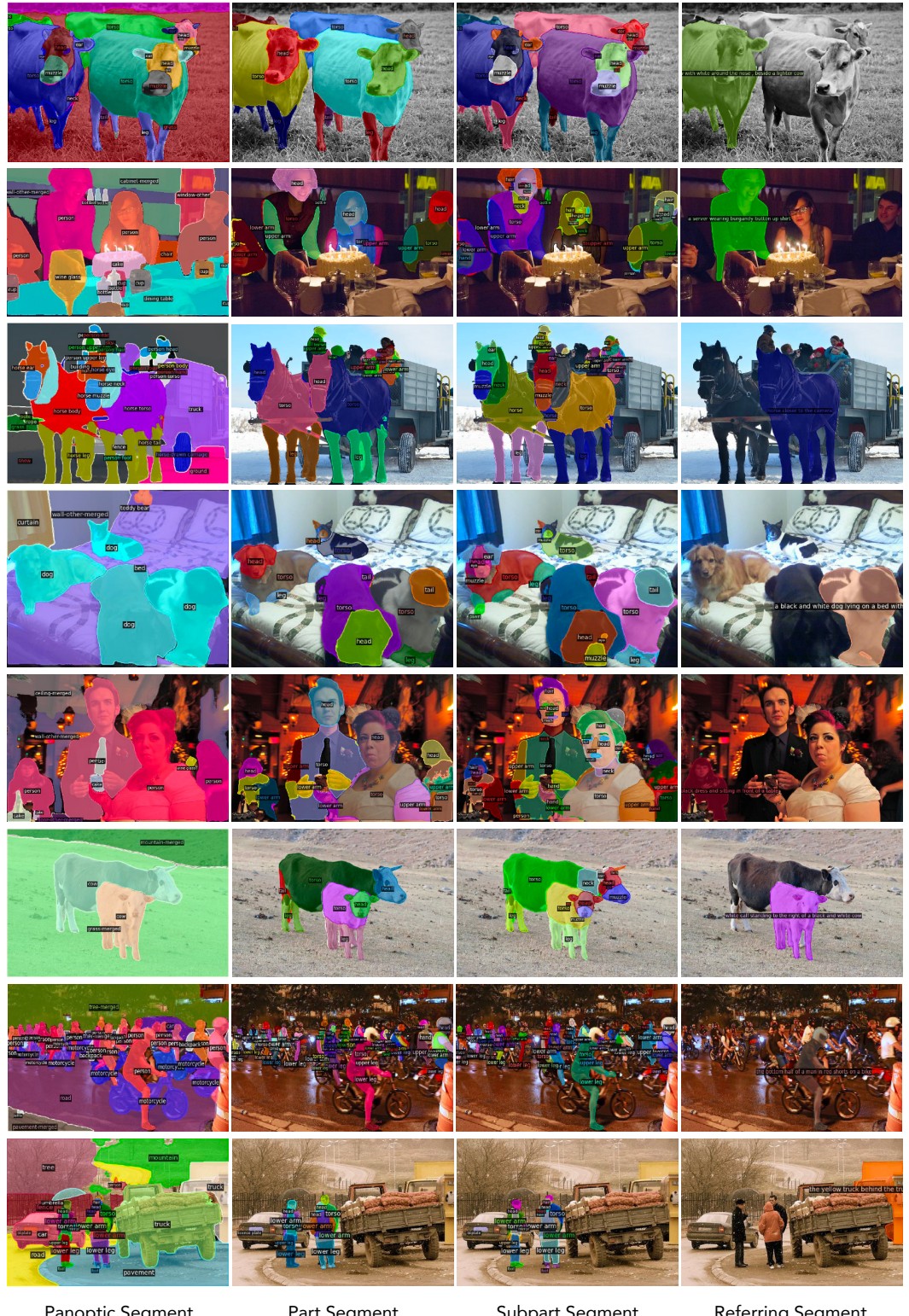

Panoptic Segment     Part Segment     Subpart Segment     Referring Segment