# OpenReview forum: "Hierarchical Open-vocabulary Universal Image Segmentation"
_NeurIPS.cc/2023/Conference — NeurIPS 2023 poster_

### Official Review · Reviewer_7gze · 2023-06-26

**Soundness:** 2 fair
**Presentation:** 3 good
**Contribution:** 2 fair
**Rating:** 5
**Confidence:** 4

**Summary:**

In this paper, the authors target  open-vocabulary setting and propose a universal framework for open-vocabulary semantic/instance/panoptic segmentation.  The whole framework is DETR-like. And to deal with the discrepancies between the thing and stuff classes, the authors ultilize independent decoders for thing and stuff classes. The authors benchmark their method on various datasets and achieve remarkable performance.

**Strengths:**

1) The idea is simple and easy to understand.
2) The results seem promising.

**Weaknesses:**

1) The idea seems incremental. Compared to UNITEXT, little new knowledge is introduced. The authors try to claim that their paper focuses on the open-vocabulary setting, while UNITEXT does not. However, they do not make it clear what prevents the extension of UNITEXT to the open-vocabulary setting, and their particular design for such setting. In fact, as noted on UNITEXT's github project homepage, the approach extends easily to open-vocabulary setting and achieves better performance than the proposed approach on the Seginw benchmark.

2) The comparison with other open-vocabulary semantic/panoptic segmentation methods is unfair. Since the authors used many datasets with much overlap with ADE20k-150 or Pascal Context 59 for training, but not in the previous methods, I suspect that the performance gain achieved by the proposed method comes from the training dataset. The authors should evaluate their method on a harder dataset (e.g., ADE20K-full) and perform an analysis of the overlap between their training dataset and the test data.

**Questions:**

See weakness.

**Limitations:**

NaN

---

> ### Author Rebuttal · Authors · 2023-08-09
>
> We appreciate your invaluable insights and thoughtful comments. In the following sections, we address the questions you have raised:
>
> **[W1] New knowledge introduced to UNINEXT and performance comparison with UNINEXT on open-vocabulary and part-segmentation benchmarks** \
> We thank the reviewer for taking note of the open-vocabulary capabilities of UNINEXT, which was released after our paper submission deadline. However, it's crucial to highlight that despite UNINEXT's strong performance in instance segmentation tasks, such as SeginW, its architecture lacks the capacity to effectively execute panoptic and semantic segmentation involving stuff/background classes.
> As a result, it's incapable of tackling certain common open-vocabulary benchmarks like ADE-150 panoptic segmentation and ADE-Full semantic segmentation. In Table 4 of our main paper, we highlight the substantial performance gains achieved in closed-set scenarios.
> Moreover, in the table provided below (please refer to Table R1 for comprehensive results), we present the evaluation outcomes of UNINEXT when applied naively to ADE-150, ADE-Full and CTX-459 and SeginW benchmarks. Our approach surpasses UNINEXT in both AP, which assesses instance capability, PQ, which evaluates panoptic capability, and mIoU, which evaluates the semantic segmentation performance.
> |Method| Train Data | A-150 (PQ) | A-150 (APmask) | A-150 (APbox) | A-150 (mIoU) | A-847 (mIoU) | CTX-459 (mIoU) | SeginW (APmask) |
> | --- | --- | --- | --- | --- | --- | --- | --- | --- |
> | UNINEXT (H) | O365,COCO, RefCOCO | 8.9 | 14.9 | 11.9 | 6.4 | 1.8 | 5.8 | **42.1** |
> | HIPIE (H) | O365,COCO, RefCOCO | **22.9** | **19.0** | **22.9** | **29.0** | **9.7** | **14.4** | 41.6 |
>
> Additionally, we have conducted performance comparisons between UNINEXT and HIPIE, both trained within the hierarchical segmentation settings. Our evaluation is conducted on the validation sets of two datasets: COCO for Panoptic Segmentation and PAS-P for part-segmentation.
> | Method | Train Data | COCO (PQ) | COCO (APmask) | COCO (APbox) | COCO (mIoU) | PAS-P (mioUPartS) |
> | --- | --- | --- | --- | --- | --- | --- |
> | UNINEXT (H) | O365,COCO, RefCOCO, PAS-P | 37.3 | 60.1 | 49.9 | 21.3 | 52.0 |
> | HIPIE (H) | O365,COCO, RefCOCO, PAS-P | **58.0** | **61.3** | **51.9** | **66.8** | **63.8** |
>
> In summary, we achieve competitive, if not superior, performance to UNINEXT in instance segmentation, while significantly enhancing its performance in part-segmentation, open-vocabulary panoptic and semantic segmentation—a testament to the strides we've made in advancing the model's capabilities.
>
> We sincerely appreciate your valuable suggestions and feedback. We will incorporate all of these results into our revision.
>
> **[W2] Is there any overlap between the training dataset used and ADE-150 or Pascal Context? Could you evaluate your approach on more challenging datasets, like ADE20K-full, and provide an analysis of data overlap between your training dataset and the test data?**  \
> We genuinely appreciate your insightful question, which often goes overlooked in prior research. To the best of our knowledge, we are not aware of any substantial data overlap between the datasets we employ for training and ADE-150, as well as Pascal Context. It's noteworthy that these datasets are commonly utilized in a multitude of earlier works, including OpenSeed.
>
> In direct response to your question, we have included performance metrics on more rigorous datasets like ADE-full and Pascal-Context-459 in the Table below (please also see Table R1). Our approach exhibits superior performance compared to previous works that were trained under similar settings.
>
> |Method|Venue|Dataset|A-150 (PQ)|A-150 (APmask)|A-150 (APbox)|A-150 (mIoU)|A-847 (mIoU)|CTX-459 (mIoU)|SeginW (APmask)|
> | -- | -- | -- | -- | -- | -- | -- | -- | -- | -- |
> |OpenSeed|ICCV2023|O365,COCO|19.7|15.0|17.7|23.4|-|-|36.1|
> |X-Decoder|CVPR2023|COCO,CC3M,SBU-C,VG,COCO-Caption,(Florence)|21.8|13.1|-|**29.6**|9.2|**16.1**|32.2|
> |UNINEXT|CVPR2023|O365,COCO,RefCOCO|8.9|14.9|11.9|6.4|1.8|5.8|**42.1**|
> |HIPIE (ours)|-|O365,COCO,RefCOCO|**22.9**|**19.0**|**22.9**|29.0|**9.7**|14.4|41.6|
>
> Furthermore, our model's universality empowers it to effectively harness detection datasets such as Object365—an advantage unique to our architecture. This sets us apart from models like X-Decoder, which are constrained due to their decoder design, preventing the use of bounding-box-only datasets. Similarly, other methods like [1,2,4,5] exclusively concentrate on semantic segmentation and lack instance-awareness. We argue that our universality is a strength but weakness because training on multiple tasks benefits each other.
>
> *Hope our explanation and experiments are able to address your inquiries. Please don't hesitate to reply if you have any further concerns. We will integrate all your valuable suggestions into our revision, and open-source the code! Thank you!*

---

> > ### Author Response · Authors · 2023-08-13
> > **We genuinely appreciate the time and thought you invested in reviewing our paper**
> >
> > Dear Reviewer 7gze,
> >
> > We genuinely appreciate the time and thought you invested in reviewing our paper. Your feedback has been incredibly valuable in enhancing the quality of our work.
> >
> > We're pleased to inform you that we have carefully addressed all the questions and concerns you raised in your reviews. Here's a brief summary of our actions:
> >
> > - **Clarifications on hierarchical segmentation**: We've included a ***diagram (Fig. R1 in the rebuttal PDF) to illustrate the essential differences*** from naively training the model on different granularities. Additionally, we've presented ***qualitative results in Fig. R2*** and ***quantitative results in Table R2***, effectively showcasing the benefits of our design choice.
> >
> > - **Comparison with works trained on Part-Segmentation datasets**: We've ***significantly boosted mIoUPartS by over 11.8 points*** in comparison to the UNINEXT baseline on the part-segmentation dataset (Please refer to Table R2 for detailed results).
> >
> > - **Results comparison with UNINEXT on SeginW**: [***Please note that the results of UNINEXT on SeginW were released in June 2023, which is after NeurIPS deadline.***] Despite UNINEXT's strength in instance segmentation tasks, such as SeginW, ***UNINEXT's architecture lacks the capacity to execute panoptic and semantic segmentation*** effectively. In Table R1, we present the evaluation outcomes of UNINEXT when applied naively to ADE-150, ADE-Full, CTX-459, and SeginW benchmarks. ***Our approach outperforms UNINEXT by a large margin in instance segmentation, panoptic segmentation, and semantic segmentation.***
> >
> > - **Overlap between training dataset and ADE-150 or Pascal Context**: We want to highlight that ***these datasets are commonly used in earlier works***, including OpenSeed. Additionally, we ***are not aware of any significant data overlap*** between the datasets we use for training and ADE-150, as well as Pascal Context.
> >
> > Your insights have significantly contributed to refining our work, and we believe the paper has greatly benefited from your expertise.
> >
> > ***If you have any further questions or suggestions, please don't hesitate to reach out!!*** We're eager to provide any additional clarifications needed! We look forward to continuing discussions that will enrich the revision of our paper.
> >
> > Warm regards,
> >
> > Paper 384 Authors

---

### Official Review · Reviewer_4C24 · 2023-07-04

**Soundness:** 3 good
**Presentation:** 2 fair
**Contribution:** 2 fair
**Rating:** 5
**Confidence:** 3

**Summary:**

This paper presents HIPIE, an open vocabulary image segmentation model that produces segmentation from text prompts. The authors propose to decouple the segmentation of “thing” and “stuff” due to the differences in their semantic and geometric properties. By training on an additional part-level dataset, HIPIE can also perform part-level segmentation. The authors perform experiments on several open-vocabulary segmentation and referring image segmentation datasets and achieve better performances than previous state-of-the-art methods.

**Strengths:**

- The decoupling of thing and stuff decoding makes sense because of the different feature distributions between the thing classes and the stuff classes. The authors have also experimentally verified that decoupling helps in quantitative measures.

- The proposed method has strong performance on several popular datasets including COCO panoptic segmentation, ADE20K, the referring COCO dataset, etc.

- The authors show that the proposed method can also work on part-level segmentation after being trained on part segmentation datasets. They show that the proposed method works better than Grounding DINO on at least one example in part segmentation which is an interesting application.


**Weaknesses:**

- The paper claims a hierarchical representation which I find weak. There is only a small paragraph (Section 3.7) about hierarchical segmentation, and it is about part-level segmentation. From the text, it seems like the hierarchy is embedded in the text prompt (e.g., a head consists of ears, hair, etc.) – which any model that works on a text prompt is already capable of incorporating. The proposed model itself is not hierarchical. One thing that the proposed model does while prior work doesn’t is that it trains on part segmentation dataset (Pascal-Panoptic-Parts). However, it seems that the authors have only tested part segmentation performance on the same dataset, hence not demonstrating the “open vocabulary” capability. Overall, I find the claim of “hierarchical model” as a contribution confusing.

- Most of the performance improvements seem to come from decoupling. It appears to me that decoupling leads to extra model parameters and run-time because there are two decoders to run for each input image. There are no comparisons with recent models in these two regards.

- It is unclear why “feature fusion” should not be performed for the “stuff” branch.


**Questions:**

Table 3 – what is the difference between the top part and the bottom part?

**Limitations:**

The authors present a section for limitations (Section A5) in the supplementary material. It points to several future work directions rather than limitations of the proposed model. One potential limitation is that the part-level segmentation might not generalize to vocabulary beyond the training set.

---

> ### Author Rebuttal · Authors · 2023-08-09
>
> We appreciate your invaluable insights and thoughtful comments. In the following sections, we address the questions you have raised:
>
> **[W1] How does the hierarchical segmentation process occur in the proposed model? Is the model inherently hierarchical in its architecture? How to demonstrate the “open vocabulary” capability for part-segmentation?**
>
> Nice questions! Sorry for the confusion! We have indeed integrated unique designs tailored to hierarchical segmentation. In our efforts to elucidate the pivotal differences from previous methods and from naively training the model on part-segmentation datasets, we have included illustrative diagrams in the rebuttal PDF.
>
> Specifically, we concatenate class names from various hierarchical levels and contrast a mask embedding with these labels within the training loss. To illustrate, consider the example of "person head", we establish positive targets for both "person" and "head" individually, while designating negative targets for all other class names. This approach starkly contrasts with the outcomes of naively applying alternative methods, where "person head" might unintentionally garner negative targets from classes like "person body" or "person eye."
> Instead of treating each class name as an ordinary multi-word class label, our design uniquely captures the hierarchical nature of the underlying semantics. At inference, we run the same image once for each level of hierarchy and combine the final outputs.
>
> ***In Figure R1***, we visually articulate the design disparities when compared to methods like UNINEXT and ODISE.
>
> ***In Figure R2***, we show that our design benefits open-vocabulary settings and allows zero-shot inference for object parts on novel concepts.
>
> Additionally, we have conducted performance comparisons between UNINEXT and HIPIE, both trained within the hierarchical segmentation settings. Our evaluation is conducted on the val sets of two datasets: COCO for Panoptic Segmentation and PAS-P for Part-Segmentation.
>
> |Method|Train Data|COCO (PQ)|COCO(APmask)|COCO (APbox)|COCO (mIoU)|PAS-P (mioUPartS)|
> |-|-|-|-|-|-|-|
> |UNINEXT (H)|O365,COCO,RefCOCO,PAS-P|37.3|60.1|49.9|21.3|52.0|
> |HIPIE (H)|O365,COCO,RefCOCO,PAS-P|**58.0**|**61.3**|**51.9**|**66.8**|**63.8**|
>
> In the context of open-vocabulary part-segmentation, we acknowledge that a quantitative assessment of the open vocabulary capability is hindered by the limited availability of part-segmentation datasets, where most datasets have similar classes. However, we provide qualitative analysis in Figure R2, where we showed that our design benefits open-vocabulary settings and allows zero-shot inference of parts on novel objects. We are committed to further advancing our research in open-vocabulary part-segmentation. In future endeavors, we plan to label a new dataset and release it for the evaluation of open-vocabulary part-segmentation.
>
> **[W2] It appears to me that decoupling leads to extra model parameters and run-time because there are two decoders to run for each input image. There are no comparisons with recent models in these two regards.**
>
> Another nice question! Compared with UNINEXT, we introduce 30M more parameters, or 4% increase in total parameters (805M vs 775M). In terms of inference speed, our model leads to a small performance loss on A100 (1.31s vs 1.42s per iteration). However, given the observed performance gain and new task capabilities, we believe such cost is justifiable.
>
> **[W3] It is unclear why “feature fusion” should not be performed for the “stuff” branch**
>
> Regarding the decision to refrain from "feature fusion" within the "stuff" branch, we would like to reference the analysis detailed in Section 1 (Lines 35-64) and Figure 2 of our paper. Specifically, in Figure 2, we observed that: 1) Noticeable discrepancies exist in the between class similarities of textual and visual features between stuff and thing classes. 2) Stuff classes exhibit significantly higher levels of similarity in text features than things. This observation suggests that integrating textual features may yield more significant benefits in generating discriminative features for thing classes compared to stuff classes. Consequently, for thing classes, we adopt an early image-text fusion approach to fully leverage the benefits of discriminative textual features.
>
> Furthermore, we have conducted empirical validation on MSCOCO, as presented in the table (copied from Table 4 in our paper), which underscores the superiority of our design in comparison to alternative design choices.
>
> | Method | PQ | APmask | mIoU |
> | ------ | --------- | --------- | --------- |
> | Baseline - Fig. 4a | 44.6 | 42.5 | 66.8 |
> | Decoupled (Fusion: Stuff + Things) - Fig. 4b | 50.0 | **44.4** | 77.1 |
> | Decoupled (Fusion: Things) - Fig. 4c   | **51.3** | **44.4** | **77.3** |
>
> *Hope our explanation and experiments are able to address your inquiries. Please don't hesitate to reply if you have any further concerns. We will integrate all your valuable suggestions into our revision, and open-source the code! Thank you!*

---

> > ### Author Response · Authors · 2023-08-13
> > **We truly appreciate your dedicated time and thoughtful review of our paper**
> >
> > Dear Reviewer 4C24,
> >
> > We truly appreciate your dedicated time and thoughtful review of our paper. Your feedback has been immensely valuable in enhancing the quality of our work.
> >
> > We're pleased to share that we have thoroughly addressed all the questions and concerns you raised in your reviews. Here's a concise summary of our actions:
> >
> > - **Clarifications on hierarchical segmentation**: We've included **a diagram (Fig. R1 in the rebuttal PDF) to illustrate the essential differences** from naively training the model on different granularities. Moreover, we've presented ***qualitative results in Fig. R2*** and ***quantitative results in Table R2***, effectively showcasing the benefits of our design choice.
> >
> > - **Comparison with works trained on Part-Segmentation datasets**: We've significantly ***boosted mIoUPartS by over 11.8 points*** in comparison to the UNINEXT baseline on the part-segmentation dataset (Please refer to Table R2 for detailed results).
> >
> > - **Impact of decoupling on model parameters and run-time**: We've addressed this concern by highlighting that ***the total parameters increased by only 4%*** compared to UNINEXT (805M vs 775M), leading to ***substantial performance gains*** and the ***introduction of new task capabilities***.
> >
> > Additionally, we've attended to other minor questions and aspects as needed.
> >
> > Your insights have greatly contributed to refining our work, and we believe the paper has significantly benefited from your expertise.
> >
> > ***If you have any further questions or suggestions, please don't hesitate to reach out!!!*** We're eager to provide any additional clarification needed! We look forward to continuing discussions that will enrich the revision of our paper.
> >
> > Warm regards,
> >
> > Paper 384 Authors

---

> > > ### Comment · Reviewer_4C24 · 2023-08-13
> > >
> > > I thank the authors for the detailed response.
> > >
> > > 1. The authors explained how hierarchical modeling is used in their models and how it is different from prior works in the rebuttal. This distinction seems to be core to this paper's contribution as a "hierarchical model". Unless I am missing something, this distinction is entirely absent in the original submission. I think the paper should also be explicit that this is the source of the hierarchy in the model. Have the authors performed ablation studies on the use of this hierarchical textual modeling? The difference between HIPIE and UNINEXT is influenced by many other factors and is therefore not indicative of the advantage of the proposed hierarchy. I understand that this would put a heavy workload on the authors (if these ablations have not been done) at this stage, but then this important technical detail is only brought up in the rebuttal.
> > >
> > > 2. The decision to not use "feature fusion" in the stuff branch still seems largely empirical. The cited rationales suggest that the feature fusion branch helps with "things" more than "stuff" but not that it would harm the "stuff" branch. That said, I am okay with an empirical contribution.

---

> > > > ### Author Response · Authors · 2023-08-14
> > > > **Replying to Official Comment by Reviewer 4C24**
> > > >
> > > > Dear Reviewer,
> > > >
> > > > We are pleased that our rebuttal has been able to address some of your previous questions. We are now ready to provide answers to your new questions:
> > > >
> > > > - **Additional Ablation Study on Hierarchical Textual Embedding**: We have previously conducted a small-scale ablation study on the hierarchical scheme using the HIPIE model with a ResNet50 backbone. We started with the HIPIE model trained on the COCO-Panoptic dataset using CLIP and BERT backbones as the base (details in Table A4 in appendix), and then performed additional training on a part segmentation dataset. The notation HIPIE (Fig. R1 a), (Fig. R1 b), and (Fig. R1 c) corresponds to the left, center, and right diagrams shown in Figure R1 in the rebuttal PDF. Among these, HIPIE (Fig. R1 a) achieved the best performance, with an improvement of +2.7 mIoUPartS compared to the alternative design choices, and +2.8 mIoUPartS compared to the previous state-of-the-art with a similar-sized backbone.
> > > > || Backbone |  Text Encoder |  Label Target| mIoUPartS (PAS-P) |
> > > > |--|--|--|--|--|
> > > > | HIPIE (Fig. R1 a) | R50  | BERT |Hierarchal Concatenation | **57.2** |
> > > > | HIPIE (Fig. R1 b) | R50  | BERT |Naive (UNINEXT) |54.5|
> > > > | HIPIE (Fig. R1 c) | R50  | CLIP |Naive (ODISE) | 48.7|
> > > > | JPPF [22] | EffNet-B5 | - | Close Set| 54.4 |
> > > >
> > > > We appreciate your suggestion on the importance of highlighting the hierarchical segmentation aspect and providing more comprehensive details in the main paper. We are fully committed to incorporating all these details in our rebuttal into the revision!
> > > >
> > > > - **The decision to not use "feature fusion" in the stuff branch**: Nice question! We agree that the design choices indeed backed by empirical experiments. However, we want to kindly emphasize that the decision to exclude "feature fusion" in the stuff branch stems from a deeper analysis. In Figure 2 of the introduction section, we conducted a statistical study of the disparities in between-class similarities of visual and textual features between stuff and thing classes. Notably, we found that ***stuff classes exhibit notably higher levels of similarity in text features*** compared to thing classes. To address this, we adopt a late image-text fusion strategy to ***counterbalance the potential negative impact of non-discriminative stuff class textual features***. Our empirical experiments provide validation for this observation. For more in-depth insights, we kindly ask you to review the details in lines 50-64. ***We will provide more comprehensive details on this aspect in the revision.***
> > > >
> > > > We genuinely value your input and hope our responses satisfactorily address your new inquiries.
> > > >
> > > > ***Please do not hesitate to reply if you have further questions!*** We look forward to improving the clarity and depth of our work with your valuable input!
> > > >
> > > > Best wishes,
> > > >
> > > > Paper 384 Authors

---

> > > > > ### Comment · Reviewer_4C24 · 2023-08-14
> > > > >
> > > > > Thank you for the follow-up. These additions help the paper and I am raising the score to a borderline accept.

---

> > > > > > ### Author Response · Authors · 2023-08-14
> > > > > > **We sincerely appreciate your decision to upgrade the score!!!**
> > > > > >
> > > > > > Dear Reviewer,
> > > > > >
> > > > > > We sincerely appreciate your decision to upgrade the score to a borderline accept! Your insightful feedback will be incorporated into the revision.
> > > > > >
> > > > > > Wishing you a wonderful day!
> > > > > >
> > > > > > Paper 384 Authors

---

### Official Review · Reviewer_EcWi · 2023-07-06

**Soundness:** 2 fair
**Presentation:** 3 good
**Contribution:** 2 fair
**Rating:** 4
**Confidence:** 4

**Summary:**

The paper proposes a unified method for open-vocabulary universal image segmentation and detection methods. A text-image fusion module takes both the image features and text features and then sends the fused results to the decoder. Several designed choices are presented and compared here. The model utilizes the dataset of Objects365, COCO, RefCOCO, RefCOCOg and RefCOCO+ for training and then test on different benchmarks.

**Strengths:**

1. The overall method is clear and easy to understand.
2. The motivation to unify all the open-vocabulary segmentation and detection tasks is good.

**Weaknesses:**

1. The model is first pretrained on Object365 which has over 600K well-labeled images and then finetuned on COCO, RefCOCO, RefCOCOg and RefCOCO+. Previous open-vocabulary image segmentation methods like [1-3] only trains on COCO dataset. I don't think the comparison is fair as the model has seen much more well-labeled datasets.
2. Since the hierarchical segmentation only requires the change of text prompts, I think all previous work can also perform on this task. Maybe the authors need to tone down the hierarchical thing and add some comparison with previous work.
3. The paper is titled open-vocabulary, however, lots of results in the paper are actually not such as the COCO results in Table 2, 5, 6. It's ok to report the results, but the authors need to report the open-vocabulary results which should be the main focus.
4. Some important comparisons are missing [4, 5] on open-vocabulary semantic segmentation.

[1] Scaling Open-Vocabulary Image Segmentation with Image-Level Labels.\
[2] Open-vocabulary panoptic segmentation with maskclip.\
[3] Open-vocabulary panoptic segmentation with text-to-image diffusion models.\
[4] Open-Vocabulary Semantic Segmentation with Mask-adapted CLIP.\
[5] Side Adapter Network for Open-Vocabulary Semantic Segmentation.

**Questions:**

Table 3 seems confusing, why do the authors split the table into two parts?

**Limitations:**

Yes.

---

> ### Author Rebuttal · Authors · 2023-08-09
>
> Thank you for your insightful inquiries, and we will provide detailed responses to each of them below:
>
> **[W1] Concern about HIPIE's pre-training on Object365, which is unfair.**
>
> With regard to the use of large datasets, we'd like to highlight that our research is centered on universal models capable of addressing multiple tasks within a unified framework. As a result, we've trained our model on various datasets that collectively support these target tasks. This approach is aligned with recent endeavors such as OpenSeed (ICCV2023), GroundingDINO (arXiv2023), and X-Decoder (CVPR2023), which actually have used datasets of considerably larger scales than ours.
>
> In the table provided below, we present a comprehensive comparison against con-current works that have been trained under comparable settings. Importantly, our work shows a remarkable performance superiority and obtains significant outperformance of the UNINEXT baseline—a model that also holds universal capabilities.
>
> |Method|Venue|Dataset|A-150 (PQ)|A-150 (APmask)|A-150 (APbox)|A-150 (mIoU)|A-847 (mIoU)|CTX-459 (mIoU)|SeginW (APmask)|
> | -- | -- | -- | -- | -- | -- | -- | -- | -- | -- |
> |OpenSeed|ICCV2023|O365,COCO|19.7|15.0|17.7|23.4|-|-|36.1|
> |X-Decoder|CVPR2023|COCO,CC3M,SBU-C,VG,COCO-Caption,(Florence)|21.8|13.1|-|**29.6**|9.2|**16.1**|32.2|
> |UNINEXT|CVPR2023| O365,COCO,RefCOCO|8.9|14.9|11.9|6.4|1.8|5.8|42.1|
> |HIPIE (ours)|-| O365,COCO,RefCOCO|**22.9**|**19.0**|**22.9**|29.0|**9.7**|14.4|**41.6**|
>
> In contrast to prior open-vocabulary segmentation methods, our HIPIE harnesses a unique advantage through the universality inherent in our model, which empowers us to effectively exploit both segmentation and detection datasets such as Object365. Prior works such as [3] and X-Decoder cannot use bounding-boxes-only datasets because of their decoder design. Other works such as [1,2,4,5] focus on semantic segmentation and do not have instance awareness.
>
> **[W2] Maybe the authors need to tone down the hierarchical thing and add some comparison with previous work.**
>
> Sorry for the confusion! We have indeed integrated unique designs tailored to hierarchical segmentation. In our efforts to elucidate the pivotal differences from previous methods and from naively training the model on part-segmentation datasets, we have included illustrative diagrams in the rebuttal PDF. Specifically, we concatenate class names from various hierarchical levels and contrast a mask embedding with these labels within the training loss. To illustrate, consider the example of "person head", we establish positive targets for both "person" and "head" individually, while designating negative targets for all other class names. This approach starkly contrasts with the outcomes of naively applying alternative methods, where "person head" might unintentionally garner negative targets from classes like "person body" or "person eye." Instead of treating each class name as an ordinary multi-word class label, our design uniquely captures the hierarchical nature of the underlying semantics. At inference, we run the same image once for each level of hierarchy and combine the final outputs.
>
> **In Figure R1**, we visually articulate the design disparities when compared to methods like UNINEXT and ODISE.
>
> **In Figure R2**, we show that our design benefits open-vocabulary settings and allows zero-shot inference for object parts on novel concepts.
>
> Additionally, we have conducted performance comparisons between UNINEXT and HIPIE, both trained within the hierarchical segmentation settings. Our evaluation is conducted on the val sets of two datasets: COCO for Panoptic Segmentation and PAS-P for Part-Segmentation.
>
> |Method|Train Data|COCO (PQ)|COCO(APmask)|COCO (APbox)|COCO (mIoU)|PAS-P (mioUPartS)|
> |-|-|-|-|-|-|-|
> |UNINEXT (H)|O365,COCO,RefCOCO,PAS-P|37.3|60.1|49.9|21.3|52.0|
> |HIPIE (H)|O365,COCO,RefCOCO,PAS-P|**58.0**|**61.3**|**51.9**|**66.8**|**63.8**|
>
> **[W3] Open-vocabulary results should be the main focus**
>
> We thank the reviewer for pointing out the importance of open-vocabulary performance. The closed-set results shown in Table 2, 5, and 6 primarily underscore the universality of our model, a core contribution we emphasize. This universal capability is particularly noteworthy, considering that many prior works like X-Decoder and ODISE lack the capacity to perform detection and referring expression comprehension due to their decoder designs.
>
> In addition to the open-vocabulary results reported in the main paper, we also provide more results on Object Detection in the Wild (ODinW) and Segmentation in the Wild (SeginW) in our appendix and we obtain significantly better results. Furthermore, we provide more open-vocabulary results in the first Table (rebuttal PDF) with ADE-full and Pascal-Context-459 featuring many novel classes. Our model achieves competitive performance in these datasets as well.
>
> **[W4] Comparisons with [4, 5] are missing on open-vocabulary semantic segmentation.**
>
> We will add it in the revision! We can outperform [4] (+0.7 mIoU on ADE-full and + 2.1 mIoU on CTX-459) and we do not surpass [5]. However, it's important to highlight that these models are designed specifically for semantic segmentation, whereas our model's capabilities encompass detection, instance segment, referring segment, and part segmentation as well. Also, these models rely on the COCO-Stuff dataset, which has more stuff classes compared to the COCO-Panoptic dataset that we employ, while our current setting is more favorable for object detection and instance segmentation. We intend to explore the implications of the COCO-Stuff dataset in the revision.
>
> **[Questions]** The first part of Table 3 presents a comparison with methods using RN50 as their backbone, while the second part lists models utilizing larger backbones. We will fix it in the revision.
>
> *Hope our explanation and experiments can address your inquiries. We will integrate all your valuable comments into our revision!*

---

> > ### Author Response · Authors · 2023-08-13
> > **We sincerely thank you for your time and effort invested in reviewing our paper**
> >
> > Dear Reviewer EcWi,
> >
> > We sincerely thank you for your time and effort invested in reviewing our paper. Your feedback has proven to be invaluable in elevating the quality of our work.
> >
> > We're delighted to inform you that we've diligently addressed each of the questions and concerns you raised in your reviews. Here's a brief summary of our actions:
> >
> > - **HIPIE's pre-training on Object365**: To ensure a fair comparison, we've conducted a comprehensive evaluation (Table R1) against con-current works, including OpenSeed (ICCV2023), GroundingDINO (arXiv2023), and X-Decoder (CVPR2023), which employ datasets of notably larger scales than ours. We're thrilled to report that our HIPIE model ***outperforms not only the UNINEXT baseline but also these con-current works***, showcasing its significant capabilities.
> >
> > - **Clarifications on hierarchical segmentation**: To enhance clarity, we've included ***a diagram (Fig. R1 in the rebuttal PDF) illustrating the key distinctions*** from naively training the model on different granularities. Additionally, we've provided ***qualitative results in Fig. R2*** and ***quantitative results in Table R2***, demonstrating the advantages of our design choice.
> >
> > - **Comparing with UNINEXT trained on Part-Segmentation datasets**: We've significantly ***improved mIoUPartS by over 11.8 points*** compared to the UNINEXT baseline on the part-segmentation dataset. (Please see Table R2 for details)
> >
> > - **More open-vocabulary results**: In addition to the extensive evaluation results (***over 40 datasets***) presented in our main paper and appendix, we've included ***additional results on A-150, A-847, CTX-459, and SeginW*** in the rebuttal PDF. Notably, we've achieved state-of-the-art results on most of these benchmarks, highlighting our model's competitiveness, even when compared to con-current works.
> >
> > Furthermore, we're committed to addressing other minor questions related to table design.
> >
> > Your input has been instrumental in shaping our revisions, and we're confident that the paper has greatly benefited from your expertise.
> >
> > ***If you have any further questions or suggestions, please don't hesitate to share them!!!*** We will be happy to answer them! We eagerly anticipate further discussions that will enrich the revision of our paper.
> >
> > Warm regards,
> >
> > Paper 384 Authors

---

### Official Review · Reviewer_YNZH · 2023-07-20

**Soundness:** 3 good
**Presentation:** 3 good
**Contribution:** 3 good
**Rating:** 6
**Confidence:** 3

**Summary:**

The paper propose to disentangle the representation learning and decoding for things and stuff and unify multiple segmentation tasks with different granularity (whole, part, subpart) and text formulation (reference text or category only). Extensive experiments are carried out to validate its effectiveness.

**Strengths:**

1. The paper proposed to unifying different segmentation tasks and benchmark them, which is a promissing direction for future researches.

2. The decoupling of things and stuffs in representation learning are intuitive.

3. The experimental results are extensive and promissing.

**Weaknesses:**

1. As the model only uses fully supervised data for supervision without large-scale image-text pretraining, whether its open-vocabulary capability is enough to be claimed as a open-vocabulary model? I am looking forward to see more open-vocabulary results that a quite different with exisitng training vocabulary with and without the assistance of CLIP for inference.

2.  It seems that the architecture designs (decoupling thing and stuff) of this work do not feasible to the whole/part/subpart seg. If previous methods such as UNINEXT, X-Decoder, SEEM  are also trained with part segmentation dataset (e.g., Pascal-Panoptic-Part). Can it already provide a solid results for such unified segmentation? Better with some experimental results.

**Questions:**

Generally, I appreciate the setting of this work, but still have some concerns as detailed in the weakness part. If the author can address my concern, I am willing to upgrade my rating.

Besides, I hope the author can have a detailed introduction about how to do the hierarchical segmentation in 3.7, you need to do the segmentation multiple times or just once? If the text query has both whole, part and subpart categories, how will the model do the inference.

---

> ### Author Rebuttal · Authors · 2023-08-09
>
> We appreciate your invaluable insights and thoughtful comments. In the following sections, we address the questions you have raised:
>
> **[W1] Open-vocabulary results with and without the assistance of CLIP for inference**
>
> Nice question! We sincerely appreciate the reviewer for highlighting the absence of extensive large-scale image-text pre-training. This gap indeed serves as one of the motivations behind our integration of CLIP assistance (it is also a common practice).
>
> To answer your question in full, we argue that the extensive labels in detection dataset and language expression in RefCOCO/+/g can also help the model align the representations of image and texts. This alignment, in turn, contributes to our model's capability to achieve open vocabulary segmentation without using CLIP.
>
> We provide additional results in the table below. In comparison to other universal models, our model, when integrated with CLIP, achieves the state-of-the-art performance in ADE-full benchmark with 847 classes. Moreover, the performance exhibited by HIPIE, even in the absence of CLIP, surpasses the UNINEXT baseline by a significant margin. This outcome underscores our model's commendable open vocabulary capability, even without reliance on CLIP assistance.
>
> |Method|Venue|Dataset|A-150 (PQ)|A-150 (APmask)|A-150 (APbox)|A-150 (mIoU)|A-847 (mIoU)|CTX-459 (mIoU)|SeginW (APmask)|
> | -- | -- | -- | -- | -- | -- | -- | -- | -- | -- |
> |OpenSeed|ICCV2023|O365,COCO|19.7|15.0|17.7|23.4|-|-|36.1|
> |X-Decoder|CVPR2023|COCO,CC3M,SBU-C,VG,COCO-Caption,(Florence)|21.8|13.1|-|**29.6**|9.2|**16.1**|32.2|
> |UNINEXT|CVPR2023|O365,COCO,RefCOCO|8.9|14.9|11.9|6.4|1.8|5.8|42.1|
> |HIPIE w/o CLIP (ours)|-|O365,COCO,RefCOCO|18.1|16.7|20.2|19.8|4.8|12.2|41.0|
> |HIPIE w/ CLIP (ours)|-|O365,COCO,RefCOCO, (CLIP) |**22.9**|**19.0**|**22.9**|29.0|**9.7**|14.4|**41.6**|
>
> **[W2 and Questions] How to do hierarchical segmentation? If previous methods such as UNINEXT, X-Decoder, SEEM are also trained with part segmentation dataset (e.g., Pascal-Panoptic-Part). Can it already provide solid results for such unified segmentation?**
>
> We are sorry for the confusion caused by insufficient architectural description. We have indeed incorporated distinct designs specifically tailored for hierarchical segmentation. In our efforts to elucidate the pivotal differences from previous methods and from naively training the model on part-segmentation datasets, we have included illustrative diagrams in the rebuttal PDF.
>
> Specifically, we concatenate class names from various hierarchical levels and contrast a mask embedding with these labels within the training loss. To illustrate, consider the example of "person head", we establish positive targets for both "person" and "head" individually, while designating negative targets for all other class names. This approach starkly contrasts with the outcomes of naively applying alternative methods, where "person head" might unintentionally garner negative targets from classes like "person body" or "person eye."
> Instead of treating each class name as an ordinary multi-word class label, our design uniquely captures the hierarchical nature of the underlying semantics. At inference, we run the same image once for each level of hierarchy and combine the final outputs.
>
> In Figure R1, we visually articulate the design disparities when compared to methods like UNINEXT and ODISE.
>
> In Figure R2, we show that our design benefits open-vocabulary settings and allows zero-shot inference for object parts on novel concepts.
>
> Additionally, we have conducted performance comparisons between UNINEXT and HIPIE, both trained within the hierarchical segmentation settings. Our evaluation is conducted on the val sets of two datasets: COCO for Panoptic Segmentation and PAS-P for Part-Segmentation.
>
> |Method|Train Data|COCO (PQ)|COCO(APmask)|COCO (APbox)|COCO (mIoU)|PAS-P (mioUPartS)|
> |-|-|-|-|-|-|-|
> |UNINEXT (H)|O365,COCO,RefCOCO,PAS-P|37.3|60.1|49.9|21.3|52.0|
> |HIPIE (H)|O365,COCO,RefCOCO,PAS-P|**58.0**|**61.3**|**51.9**|**66.8**|**63.8**|
>
> *Hope our explanation and experiments are able to address your inquiries. Please don't hesitate to reply if you have any further concerns. We will integrate all your valuable suggestions into our revision, and open-source the code! Thank you!*

---

> > ### Author Response · Authors · 2023-08-13
> > **We sincerely appreciate your time and effort in reviewing our paper!**
> >
> > Dear Reviewer YNZH,
> >
> > We want to express our sincere gratitude for the time and effort you've dedicated to reviewing our paper. Your feedback has been invaluable in elevating the quality of our work.
> >
> > We're pleased to inform you that we've taken great care in addressing each of the questions and concerns you raised in your reviews. Here's a summary of our actions:
> >
> > - **Open-vocabulary results without CLIP**: We're excited to report that our HIPIE model, even without CLIP, ***outperforms the UNINEXT baseline and other con-current works***, such as OpenSeeD (ICCV2023), ***by a significant margin***. The results presented in Table R1 underscore our model's remarkable open vocabulary capability, demonstrating that it stands strong even without relying on CLIP assistance.
> >
> > - **Clarifications on hierarchical segmentation**: To provide better clarity, we've incorporated ***a diagram (Fig. R1 in the rebuttal PDF) that highlights the key differences*** from naively training the model on different granularities. Additionally, we've included ***qualitative results in Fig. R2*** and ***quantitative results in Table R2***, confirming the benefits of our design choice. Particularly, we've ***achieved an impressive increase of over 11.8 points in mIoUPartS*** compared to the UNINEXT baseline on the part-segmentation dataset.
> >
> > - **More open-vocabulary results**: In addition to the extensive evaluation results (***on over 40 datasets***) we provided in our main paper and appendix, we've included ***more results on A-150, A-847, CTX-459, and SeginW*** in the rebuttal. Notably, we've achieved state-of-the-art results on most of these benchmarks, showcasing our model's competitiveness (even comparing to con-current works)!
> >
> > We genuinely appreciate your insightful input, which has greatly influenced our revisions, and we're confident that the paper has significantly improved as a result.
> >
> > ***If you have any further questions or suggestions, we welcome your input!!!*** Your continued engagement is immensely valuable, and we look forward to ongoing discussions that will further enhance the revision of our paper.
> >
> > Warm regards,
> >
> > Paper 384 Authors

---

> ### Comment · Reviewer_YNZH · 2023-08-18
>
> Thanks the explanation by authors. After reading the results in the rebuttal stage, my main concerns have been addressed.
>
> **Some suggestions**:
> 1. I think the authors should spend more effort on the paper writing for better understanding, and the explanation in rebuttal should be added into the paper during the next revision. I seems that there are similar confusions among reviewers in the pre-rebuttal version.
> 2. Besides, I encourage the author to set different subsections in the experiments part to should the *Universal*, *Open-vocabulary* and * Hierarchical* capabilities in the next revision. I believe this will help readers to better understand your contributions.
> 3. About the open-vocabulary part, I am wondering what is performance if the model is trained with more language-text paird datasets such as LION. Have authors tried that?
>
> Regarding the results provided in the rebuttal stage. I will upgrade my score to weak accept. I hope the authors could include the above-mentioned results in the next revision.

---

> > ### Author Response · Authors · 2023-08-18
> >
> > Dear Reviewer,
> >
> > We are genuinely appreciative of your decision to upgrade the score to a weak accept! Your discerning feedback will undoubtedly shape the direction of our upcoming revision. We are dedicated to carefully incorporating your suggestions to enhance the quality of the paper writing and presentation.
> >
> > Regarding your question about the model's performance when trained with additional text-image paired datasets such as LAION, we highly value this suggestion! We have solid intentions to explore this avenue in our upcoming research endeavors, and we are actively involved in conducting these experiments (utilizing larger image-text pair datasets often requires substantial training time). It's worth mentioning that our preliminary experiments have indicated the potential for incorporating LAION to enhance open-vocabulary segmentation performance, especially within the context of semantic segmentation.
> >
> > Once again, ***we extend our heartfelt appreciation for your positive feedback and great suggestions. Your decision to upgrade our score is deeply appreciated.***
> >
> > Wishing you a fantastic day and weekend ahead!
> >
> > Best regards,
> >
> > Authors of Paper 384

---

### Author Rebuttal · Authors · 2023-08-09

We extend our gratitude to the reviewers for their valuable feedback. Their insights have significantly enriched our work.
We are heartened by YNZH's recognition of our paper, where YNZH highlights the significance of our approach in *"unifying different segmentation tasks and benchmark them, which is a promising direction for future researches"*.
We appreciate EcWi's endorsement that *"the motivation to unify all the open-vocabulary segmentation and detection tasks is good"*.
We are pleased to acknowledge 4C24 and 7gze's observations that *"the proposed method has strong performance on several popular datasets"* and *"the results seem promising"*. Furthermore, we deeply appreciate 4C24's comment that *"the decoupling of thing and stuff decoding makes sense because of the different feature distributions between the thing classes and the stuff classes"*.
We will integrate all valuable suggestions into our revision, and open-source the code.

In this section, we commence by tackling the concerns that have been collectively raised. These shared concerns correspond to the three keywords in the title:

**[Hierarchical] What is the architecture design catering specifically to hierarchical segmentation and how does it compare with naively running previous methods on part segmentation dataset?**

Sorry for the confusion! We have indeed integrated unique designs tailored to hierarchical segmentation. In our efforts to elucidate the pivotal differences from previous methods and from naively training the model on part-segmentation datasets, we have included illustrative diagrams in the rebuttal PDF.

Specifically, we concatenate class names from various hierarchical levels and contrast a mask embedding with these labels within the training loss. To illustrate, consider the example of "person head", we establish positive targets for both "person" and "head" individually, while designating negative targets for all other class names. This approach starkly contrasts with the outcomes of naively applying alternative methods, where "person head" might unintentionally garner negative targets from classes like "person body" or "person eye."
Instead of treating each class name as an ordinary multi-word class label, our design uniquely captures the hierarchical nature of the underlying semantics. At inference, we run the same image once for each level of hierarchy and combine the final outputs.

***In Figure R1***, we visually articulate the design disparities when compared to methods like UNINEXT and ODISE.

***In Figure R2***, we show that our design benefits open-vocabulary settings and allows zero-shot inference for object parts on novel concepts.

***In Table R2***, we also empirically evaluated the results, which affirm our model's superior performance compared to the UNINEXT baseline trained on part datasets.

**[Universal] What novel insights or knowledge does your approach introduce when contrasted with UNINEXT and other preceding universal models?**

In this paper we consider the scope of all interesting scene understanding tasks consisting of Object Detection, Instance Segmentation, Part-segmentation, Semantic Segmentation, Panoptic Segmentation, Referring Segmentation and Referring Expression Comprehension. While there are many existing works approaching this objective, we are the first model capable of performing all these tasks.

In contrast to earlier methods, it's important to note that UNINEXT lacks the capacity for panoptic and semantic segmentation and exhibits suboptimal performance when directly applied to such tasks. X-Decoder's limitations arise from its decoder not being optimized for bounding box-based learning, rendering it unable to perform object detection. Similarly, ODISE faces constraints as it cannot conduct referring segmentation and part segmentation tasks, while also demonstrating markedly inferior results in object detection and instance segmentation. A summary of these limitations is provided in Table 1 of the main paper.

**[Open Vocabulary] More open vocabulary results are required to demonstrate the effectiveness of the method. Additionally using dataset such as O365 seems unfair when compared with previous works.**

In addition to the comprehensive evaluation results across **40** datasets presented within both the main paper and appendix materials, we have provided additional open-vocabulary results in Table R1 (see rebuttal PDF). In direct comparison to other universal models trained within similar settings, our model achieves the state-of-the-art performance in ADE-full benchmark with over 800 classes.

Compared with methods featuring segmentation-specific decoders (e.g. ODISE and X-Decoder), our model exhibits a distinctive capability – the capacity to leverage detection datasets such as Object365 for enhanced object localization. We consider this to be a strength rather than a limitation, as annotating bounding boxes is significantly simpler compared to annotating segmentation masks.

In contrast to prior methods primarily centered on open-vocabulary semantic segmentation, such as OVSeg or SAN, our model showcases a distinct capability by performing a considerably broader range of tasks with only a modest increase in the overall training dataset. All these models (OVSeg and SAN) used large scale image-text dataset such as LAION,CLIP, Florence. (Additionally, it's important to acknowledge that such a comparison might exhibit a bias towards semantic segmentation methods, mainly due to their narrower focus and common utilization of COCO-Stuff dataset, which contains a greater number of stuff classes compared to COCO-Panoptic.)

*We believe that the explanations provided above sufficiently address the primary concerns shared by the reviewers. We kindly invite you to review the individual comments for each reviewer, where you will find our responses to reviewers' specific inquiries. Your consideration is greatly appreciated. Thank you!*

---

### Decision · Program_Chairs · 2023-09-21

**Decision:**

Accept (poster)

**Comment:**

This is a system paper for hierarchical open-vocabulary universal image segmentation. The reviews are divergent, even after the discussion stage post rebuttal, with 1 weak accept, 2 borderline accepts, and 1 borderline reject.

On the positive side: 1). It's a system paper with carefully designed/adopted individual modules for image feature and text embeddings, hierarchical design, separation of the stuff and things decoding. 2). Decodings for the stuff and things have been separate, which is somewhat different from the previous approaches. 3). The hierarchical objects with the parts have been addressed.

On the negative side: 1). Even though there is a big claim about hierarchical segmentation in the title, the description about the algorithm design is nearly none in the original submission, although more explanation has been provided in the rebuttal. 2). It's a system term and there didn't appear to have existed anything that is particularly novel. 3). After all, the algorithm is still under a supervised setting, making it's open-vocabulary capability under concerns (a common point shared among reviewers). It requires to bring in the CLIP modules to enhance its open-vocabulary capability. 4). The benchmark results against the existing approaches on the open-vocabulary setting are considered somewhat unfair.

Authors have provided several tables with new results to answer the questions raised by the reviewers. Overall, these results are appreciated by the reviewers with some initial concerns resolved or partially resolved. However, concerns regarding the incremental nature of the approach and not fully-convincing experimental results remain.

There are values in the work even though it appears to be a combination of several existing approaches. Given that open-vocabulary universal segmentation is a relatively new area and a lot of results reported in the original submission and rebuttal, the work has value.